# Implicit preference for human trustworthy faces in macaque monkeys

Manuela Costa[1], Alice Gomez[1], Elodie Barat[1], Guillaume Lio[1], Jean-René Duhamel[1] & Angela Sirigu[1]

It has been shown that human judgements of trustworthiness are based on subtle processing of specific facial features. However, it is not known if this ability is a specifically human function, or whether it is shared among primates. Here we report that macaque monkeys (*Macaca Mulatta and Macaca Fascicularis)*, like humans, display a preferential attention to trustworthiness-associated facial cues in computer-generated human faces. Monkeys looked significantly longer at faces categorized a priori as trustworthy compared to untrustworthy. In addition, spatial sequential analysis of monkeys' initial saccades revealed an upward shift with attention moving to the eye region for trustworthy faces while no change was observed for the untrustworthy ones. Finally, we found significant correlations between facial width-to-height ratio– a morphometric feature that predicts trustworthiness' judgments in humans – and looking time in both species. These findings suggest the presence of common mechanisms among primates for first impression of trustworthiness.

[1] Institut des Sciences Cognitives Marc Jeannerod, CNRS, UCBL, Lyon 1, 67, boulevard Pinel, 69675 Bron, Cedex, France. These authors contributed equally: Jean-René Duhamel and Angela Sirigu.  Correspondence and requests for materials should be addressed to A.S. (email: sirigu@isc.cnrs.fr)

Trust is a fundamental psychological dimension, influencing people's willingness to cooperate[1,2], voting intentions[3], economic choices[2,4]. Trusting is taking the risk of putting one's own fate in someone else's hands, hence the importance of trustworthiness assessment to minimize this risk.

Surprisingly, research in social psychology shows that rather than being based solely on rational criteria (reputation, prior interactions), judgements of trustworthiness in humans are robustly related to specific perceptual features[5–7] such as the shape of eyebrows, cheekbones and chin. Facial features automatically capture observers' attention and lead to trustworthiness judgments after exposure to single face as brief as 33 ms, the so-called first impression effect[8]. A possible function of face first impression is to provide a sort of others' social identikit to facilitate decisions, like approaching or avoiding unfamiliar individuals, choosing a candidate during the voting process etc[9,10].

Previous studies showed that the facial width-to-height ratio (FWHR)[11], a morphometric measure of face structure, predicts explicit trustworthiness judgements[12]. In humans, faces with small FWHR are judged as more trustworthy[12]. Related results in Capuchin monkeys demonstrate a link between FWHR and assertive behavior[13], suggesting that species-typical facial traits are reliable cues used by monkeys to infer conspecifics' self-confidence. In humans, other facial attributes also predict trustworthiness judgements, including faces' femininity;[9] facial maturity[14], physical similarity to the self[15]. Todorov and colleagues further proposed the overgeneralization of emotion as the main mechanism underlying perception of trustworthiness[16], that is, neutral faces that do not display emotional expression are perceived as expressing behavioral tendencies associated with the emotion the face resemble most[10,17,18]. For instance, trustworthy and untrustworthy faces are perceived as resembling happy and angry faces, respectively[9,10,16,19].

Faces are highly salient and informative social stimuli not only in humans but in monkeys as well. It has been suggested that homologous neural and behavioral mechanisms might be involved in the processing of face cues across primate species[20–22]. Given the adaptive value of being able to infer trustworthiness in cooperative societies, one may wonder whether such a skill has an evolutionary origin. The question also arise as to whether the same mechanisms used by monkeys to analyze a conspecific's face might be recycled for making similar inferences about human faces, notably in individuals that interact extensively with humans. Here, we asked if monkeys are responsive to trustworthiness-associated human facial cues. We addressed this question by recording monkeys' and humans' eye movements using a preferential looking paradigm, a relevant approach for studying sensitivity to trustworthiness-associated cues and visual exploration strategies in both species[23].

Developmental work shows that infant rhesus monkeys exhibit innate and early experience-dependent preferences for both human and non-human primate faces[24–27]. Newborns rhesus macaques deprived from seeing their mother's or caregivers face still show preference for faces compared to objects[24]. Similarly, human newborns and foetuses look more at faces-like stimuli compared to objects[28–30] suggesting that preference for faces begins before birth and evolves during early development[31–33]. Yet, neonatal macaques may need experience to discriminate between faces and for the proper functional specialization of the visual system[34]. Remarkably, 7-month old human infants prefer to look at trustworthy faces compared to the untrustworthy ones, while such refined discrimination is not found for dominant vs submissive faces[35].

Despite evidence pointing to macaques' sophisticated abilities in processing facial features, one may still wonder why this species should be sensitive to human facial traits of trustworthiness. First, monkeys bred and raised in captivity develop considerable expertise about our physiognomy. For instance, we have shown that macaques recognize the identity of familiar humans in both face pictures and voice samples[36]. Thus, it is not unreasonable to assume that these animals learn about human trustworthiness and that they can associate these behavioral traits with human facial features. Second, like human newborns[37] baby monkeys imitate human facial movements such as tongue protrusion or lip-smacking, thus showing early abilities in reproducing human gestures[38]. Third, after observing human interactions, monkeys avoid humans who do not reciprocate in social exchanges[39,40] and they approach more humans who are imitating them[41]. Finally, monkeys and humans' visual system show strong homologies and, notably, the same temporal lobe's functional organization into multiple, hierarchically organized face patches[21,42]. Several single unit recording studies indicate that macaque specialized areas contain intermingled monkey-selective and human-selective face neurons[43–45].

In the light of such data, we reasoned that monkeys may discriminate trustworthy and untrustworthy human faces using first impression mechanisms as humans do[46]. Macaque monkeys ($N = 8$) looked at pair of faces differing in trustworthiness–associated features. We hypothesized that attention towards one of the two faces could be a sign of detection of their distinctive features. Because monkeys were not rewarded to specifically look at faces, we assumed that significantly longer looking time towards trustworthy faces may be interpreted as a preferential interest towards those stimuli.

We selected pairs of parameterized human faces ($N = 48$) drawn from Todorov et al's image database[46], each displaying the most ( $+3$ SD from baseline) and least ($-3$ SD from baseline) trustworthy version of the same facial identity. These computer-generated faces only vary on facial features that predict judgments of trustworthiness[19]. To ensure that we measured spontaneous preferences, monkeys were free to move their eyes and periodically received juice rewards to maintain gaze within the limits of the display monitor. To establish cross-species comparisons, human subjects ($N = 20$) were tested following the same procedure as in monkeys.

We show that macaque monkeys and humans look preferentially at trustworthy faces. Monkeys' visual exploration differed when attending trustworthy or untrustworthy faces. Specifically, their gaze shifted toward the eye region between the first and the second saccade only for trustworthy faces, suggesting an approach behavior toward faces bearing trust characteristics. Furthermore, in both monkeys and humans, looking times correlated significantly with FWHR of the face stimuli. These findings suggest the existence of common mechanisms among primates for first impression of trustworthiness.

## Results

**Monkeys' and humans' visual preferences**. In order to quantify gaze allocation, regions of interest (ROIs) encompassing the trustworthy and untrustworthy faces were defined. Ocular fixations within and outside these ROIs were recorded during each trial (Methods). The mean looking time was calculated as the average of the total time spent within trustworthy and untrustworthy faces for all stimulus pairs presented.

The first analysis, as expected, revealed that monkeys were attracted to both faces, spending more time on these stimuli (Akaike Information Criterion (AIC) = 6.44 .$10^5$) than predicted by a central bias model (AIC = 6.47 .$10^5$), (Fig.1) (Methods). Furthermore, monkeys discriminated between the two stimuli presented and spent significantly more time looking at trustworthy (Mean ± SD = 512.89 ± 223.87 ms) than untrustworthy

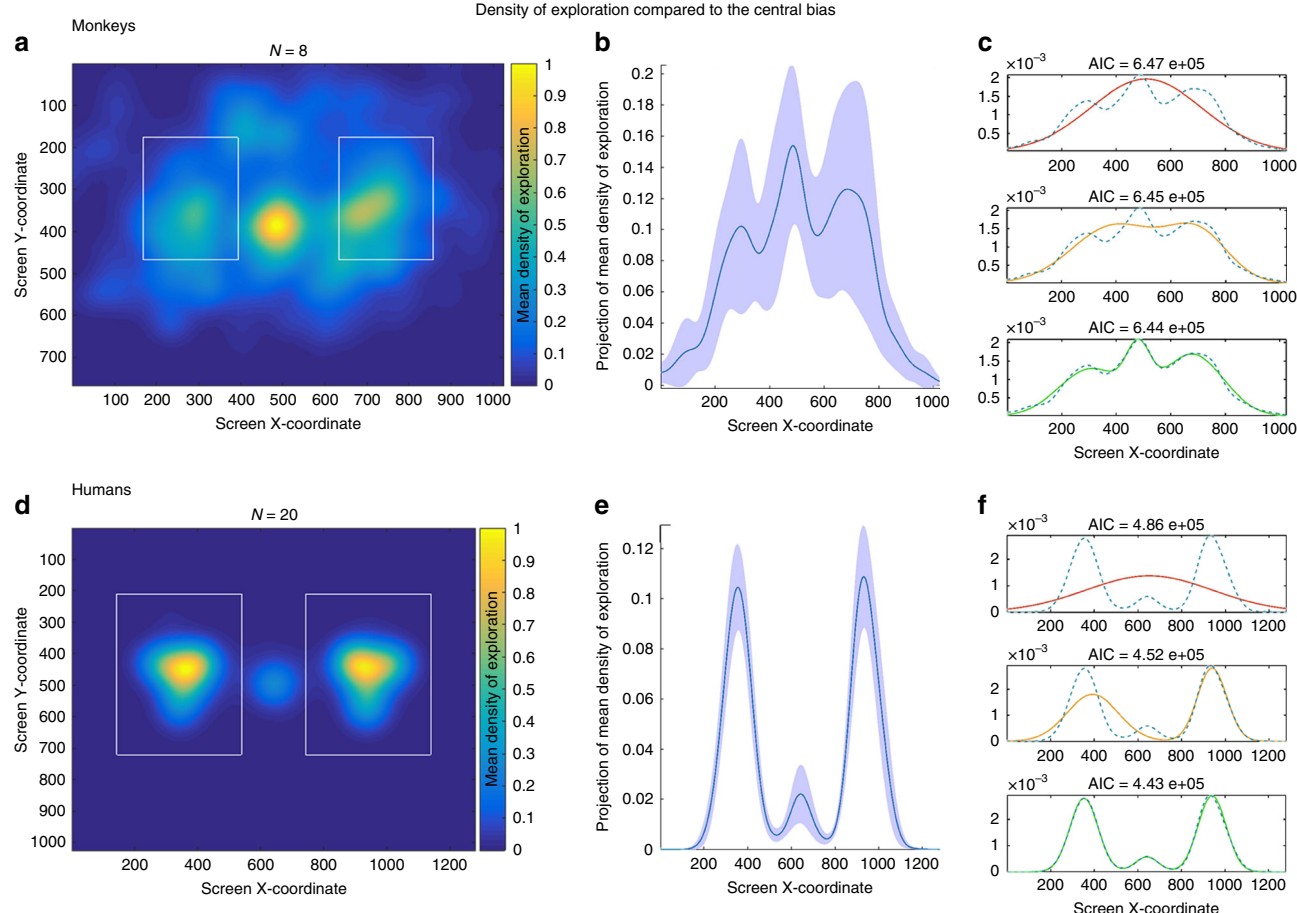

**Fig. 1** Monkeys and humans density of exploration over the screen. Left upper and lower panel: Mean density of exploration in monkeys (**a**) and humans (**d**) over the whole screen (yellow-orange indicates high density exploration, dark blue low density exploration, rectangles indicate position of faces on the screen). Middle upper and lower panel: Projection of mean density of exploration on the *x*-axis of the screen in monkeys (**b**) and humans (**e**) (with 95% confidence interval). Right upper and lower panel: AIC value according to the central bias Gaussian model (red), two faces Gaussian model (yellow), two faces and central bias Gaussian model (green) in monkeys (**c**) and humans (**f**)

(Mean ± SD = 292.60 ± 162.67 ms) faces (paired sample *t*-test; $t(7) = 3.29$; $p < 0.05$, $\eta^2 = 0.60$) (Fig.2a). This visual preference was remarkably consistent across animals, as 3 monkeys (O, Y, T) showed a highly significant bias, 1 (E) a nearly significant bias, 4 (S, V, D, Z) a non-significant positive bias toward the trustworthy faces. Importantly, none of the monkeys showed the opposite trend (Table 1).

Humans followed the same pattern, spending most of the time looking more to faces (AIC = 4.43. $10^5$) than predicted by a central bias model (AIC = 4.86 .$10^5$) (Fig.1) (Methods). Results on pupil size also indicated that humans constricted their pupil both when they gazed at trustworthy and untrustworthy faces (trustworthy: $-0.041 \pm 0.026$, $t(19) = -6.72$, $p < 0.001$; untrustworthy: $-0.038 \pm 0.02$, $t(19) = -7.16$, $p < 0.001$), thus showing attention on both stimuli (Supplementary Note 1).

Importantly, humans showed a significant bias in favor of the trustworthy ($865.35 \pm 120.44$ ms) stimuli compared to the untrustworthy ones ($796.35 \pm 105.00$ ms) (paired sample *t*-test; $t(19) = -1.87$, $p < 0.05$, $\eta2 = 0.15$) (Fig.2d). This finding was replicated in a larger sample of 54 subjects using a similar spontaneous condition and an explicit judgement of trustworthiness. Again in both cases we found a significant bias in favor of the trustworthy faces (Supplementary Note 2 and Supplementary Fig 2 & 3).

Fixations frequency analysis revealed the same significant preference in monkeys and a trend in humans. Monkeys performed more fixations over trustworthy than untrustworthy faces (trustworthy:$1.44 \pm 0.48$; untrustworthy: $0.92 \pm 0.42$; paired sample *t*-test; $t(7) = 3.24$; CI 0.14 - 0.89; $p = 0.0142$) while humans exhibited a similar trend (trustworthy faces: $2.697 \pm 0.49$; untrustworthy: $2.49 \pm 0.52$; paired sample *t*-test; $t(19) = -1.618$; $p = 0.061$).

Hence, our results reveal that both macaque monkeys and humans detected and preferred to look at human faces displaying trustworthiness-associated facial cues.

Because of this common preference across species, we explored whether monkeys and humans used similar eye gaze strategies with a focus on temporal dynamics and spatial distribution of fixations. A cluster-based permutation test (Methods) showed that, in monkeys, preference for the trustworthy faces occurred from 510 ms to 1485 ms after image onset ($p_{cluster} < 0.05$ corrected for multiple comparison) (Fig. 2b) while in humans from 200 ms ($p_{cluster} < 0.05$ corrected for multiple comparison) to 1152 ms (Fig. 2e). To provide information on the spatial distribution of visual exploration (Methods), heat maps and barycenter of eye fixations were generated (Fig. 2c–f). Overall, monkeys preferentially allocated their attention in the region surrounding the nose (Fig. 2c), while humans gazed mostly around the eye and nose regions (Fig. 2f).

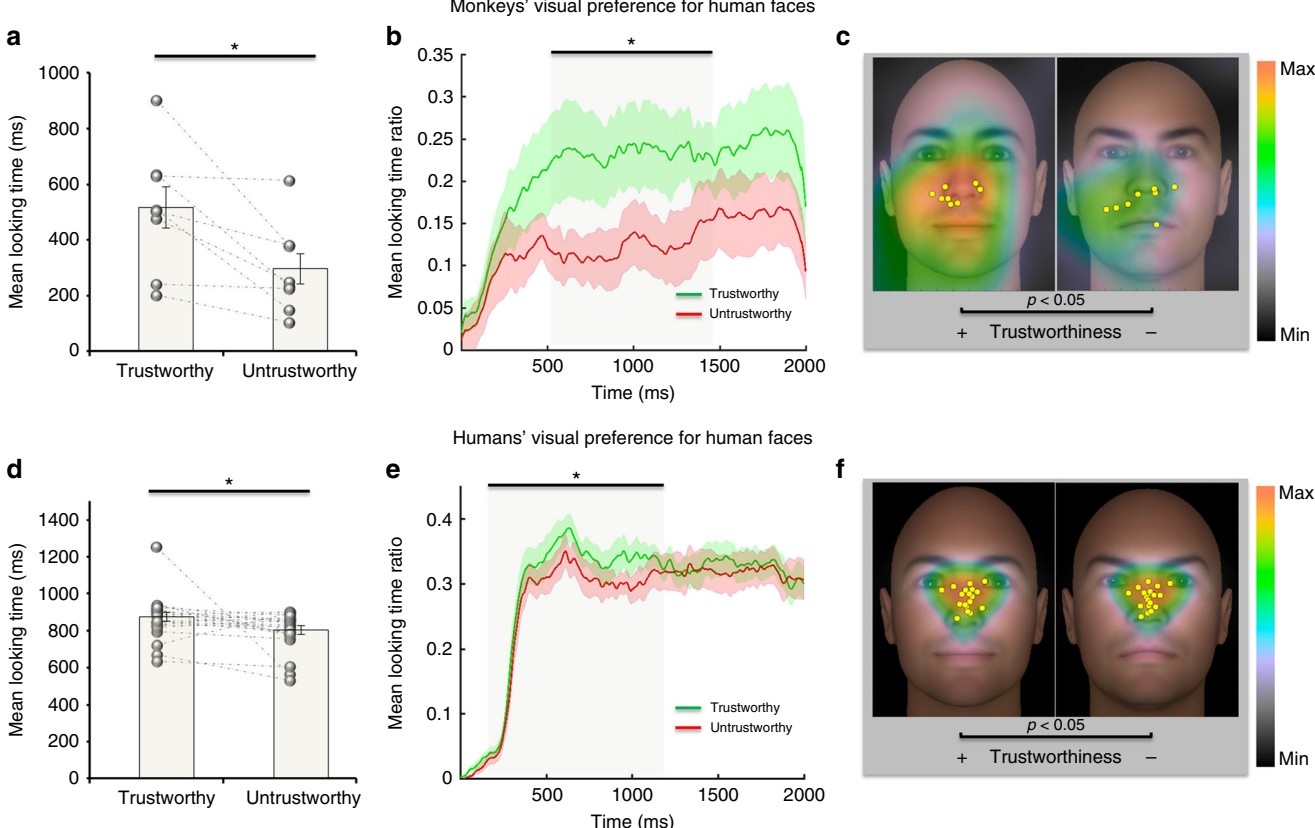

**Fig. 2** Monkeys and humans' visual preference for trustworthy faces. MONKEYS ($N = 8$): **a** Mean looking time in milliseconds (ms) for the most trustworthy (+3 SD of the neutral face) and the least trustworthy (−3SD of the neutral face) versions of the same facial identities selected from the Todorov's Database[46,65]. Circles indicate individual data points connected by dashed lines for each individual. Error bars indicate S.E.M. across 8 subjects. *$p < 0.05$, paired-sample $t$-test. Monkeys looked significantly longer at the two faces than predicted by chance and looked more at trustworthy than untrustworthy faces. **b** Time course of looking preference. Mean viewing time ratio between each facial prototype. A cluster-based permutation test showed that preference for the trustworthy faces (green line) was significant between 510 ms and 1485 ms ($P < 0.05$ corrected for multiple comparison). **c** Gaze heat maps for trustworthy and untrustworthy faces averaged across subjects (trustworthy face on the left by convention, facial prototype spatial location was counterbalanced within and between subjects). Yellow dots show fixation centers of gravity for each subject. HUMANS ($N = 20$) (**d**–**f**) Plot (**d**) show significantly longer mean looking times at trustworthy than untrustworthy faces. Error bars indicate S.E.M. across 20 subjects. Plot (**e**) show the onset of preference for trustworthy faces (200 ms to 1152 ms). Note that the average barycenter of fixation was located in the region surrounding the nose in monkeys whereas it is around the eye and nose region in humans (**c**, **f**)

**Table 1 Monkeys' individual biographical and behavioral characteristics.**

|  | Age at the time of the experiment | Sex | Species | Total looking time at the faces (ms) | Total looking time at trustworthy face (ms) | % bias for trustworthy face | p value |
|---|---|---|---|---|---|---|---|
| Y | 15 years | M | Rhesus Macaque | 657.29 | 508.6 | +27.4 | 0.004 |
| T | 6 years | M | Macaque fascicularis | 870.13 | 635.3 | +23.0 | 0.001 |
| O | 17 years | M | Rhesus Macaque | 1282.6 | 902.5 | +20.4 | 0.006 |
| S | 6 years | M | Macaque fascicularis | 308.15 | 203.2 | +15.9 | 0.18 |
| E | 6 years | M | Macaque fascicularis | 726.18 | 476.3 | +15.6 | 0.05 |
| V | 5 years | M | Rhesus Macaque | 885.2 | 503.3 | +6.9 | 0.24 |
| D | 4 years | M | Rhesus Macaque | 468.43 | 243.1 | +1.9 | 0.83 |
| Z | 13 years | F | Rhesus Macaque | 1245.95 | 630.8 | +0.6 | 0.93 |

**Spatial dynamic of eye fixations and duration of fixations.** We hypothesized that monkeys may detect human trustworthiness early after face stimuli onset. We explored monkeys' individual scanning patterns on each trial and for each stimulus type (trustworthy/untrustworthy). We computed the mean location ($x$ and $y$ coordinates) and duration of fixations for the first two fixations. We reasoned that if monkeys detect quickly the trustworthy face, their visual attention should move towards socially salient regions and therefore spatial coordinates between the first and the second fixation were expected to change. In other words, an approach behavior would be manifested by an upward shift between the first and the second fixation thus bringing the gaze close to the eye region while avoidance would keep the gaze far from the eyes area. This hypothesis is in line with previous studies showing that prolonged eye contact in great apes signals mild threat, while gaze avoidance indicates submission[47,48]. We compared across stimulus type (Trustworthy/Untrustworthy) the mean $y$ fixation coordinate (weighted by the fixation duration) of the first two fixations within the face area.

The ANOVA performed on the weighted y coordinate with the fixations (first or second) and the face type (Trustworthy vs Untrustworthy) as within subject factor revealed no main effects of face type $F(1,6) = 2.32$, $p = 0.18$ and no main effect of fixation order: $F(1,6) = 1.34$; $p = 0.29$. However, we found a significant interaction effect of stimulus type x fixation order: $F(1,6) = 8.38$; $MSE = 1014.3$; $\eta^2 = 0.58$; $p = 0.027$ (Fig. 3). Post-hoc analysis showed that the location of the first fixation was identical on both faces type (trustworthy $y$ coordinate: $355.7 \pm 13.71$; untrustworthy $y$ coordinate: $357.3 \pm 21.12$) whereas, the second fixation for the trustworthy face, was located closer to the eye region compared to the second fixation on the untrustworthy face which landed close to the mouth area (trustworthy $y$ coordinate: $335.1 \pm 10.85$; untrustworthy $y$ coordinate: $360.8 \pm 30.20$; $P = 0.02$, post hoc $t$-tests).

We also performed an ANOVA (with 2 fixations and face types) in humans. There was no main effect of the face category ($F(1,19) = 0.009$; $p = 0.92$), and no significant interaction between face category and fixation order ($F(1,19) = 0.76$; $p = 0.39$) and no main effect of fixation order ($F(1,19) = 0.69$, $p = 0.41$). This result is however not surprising given that spontaneous humans' gaze is primarily directed towards the eyes region (overall $y$ coordinate was centered on the eyes for both face type ($y$ coordinate: $460.0 \pm 5.6$).

These findings corroborate our main findings on looking time by showing that monkeys' visual exploration strategies are differently coordinated when attending trustworthy vs untrustworthy faces. Thus, results on both spatial dynamics and duration of fixations suggest an early and automatic sensitivity to trustworthiness-associated facial features in macaques.

**Correlation between monkeys' age and trust bias.** It is reasonable to assume that preference toward trustworthy-associated facial cues is shaped by experience. Therefore, we conducted an exploratory analysis where age was selected as an indicator of monkey's expertise in interacting with humans and correlated to the percentage of bias toward trustworthy-associated facial cues. The overall Pearson correlation did not reach significance ($r(8) = 0.365$ unilateral, $p = 0,187$), but when we excluded the only female outlier of the group, we found a positive Pearson correlation between age and preference for trustworthy-associated facial cues ($r(7) = 0.675$, unilateral, $p = 0.048$, without monkey Z), (Supplementary Fig.1).

**Correlation between looking time and FWHR.** In humans, FWHR–the ratio between upper facial height and the bizygomatic width - is used implicitly to form social judgements from facial appearance such as trustworthiness[12] and dominance[49,50], and we recently showed that the upper facial height is a robust predictor of trust[51].

Considering the potential importance of facial width-to-height ratio (FWHR) in human judgments of trustworthiness[12], we tested whether this morphological character might have a specific role in driving the viewing preference for trustworthiness in monkeys and humans. FWHR was calculated using standard landmarks[52]. To compute FWHR, two independent raters measured the distance between the lip and brow (upper facial height) and the left and right zygion (bizygomatic width) of each face from the entire image database. Inter-rater reliability was high for all measures (all Cronbach coefficient, $rs > 0.79$, all $ps < 0.001$). In agreement with Stirrat and Perrett's findings, we found that faces that have been judged by humans as trustworthy displayed a lower FWHR compared to the untrustworthy ones (ANOVA; $F(1,24) = 116.97$, $p < 0.05$; trustworthy: $0.02 \pm 0.019$; untrustworthy: $2.15 \pm 0.02$). The FWHR value obtained for each face was then regressed against monkeys' viewing preferences for the same face. Interestingly, total viewing time on a given face was negatively correlated to its FWHR in both monkeys (Pearson correlation, $r(48) = -0.35$, $p < 0.05$) and humans ($r(48) = -0.47$,

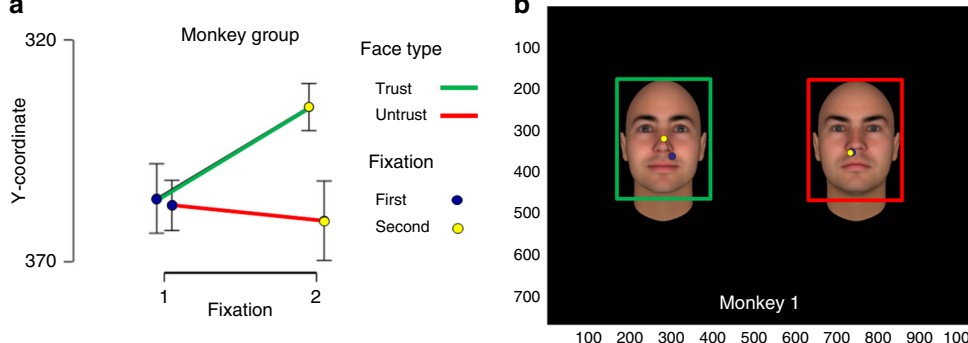

**Fig. 3** Fixations sequence analysis. **a** Graph shows monkeys' Y coordinates of first and second fixations weighted by duration (first in blue, second in yellow) and type of face (trustworthy in green, untrustworthy in red); Fixations closer to the eye region are closer to $y = 300$. The location of the first fixation on the face is not different for the trustworthy and untrustworthy face but the location of second fixation is closer to the eye region only for the trustworthy face, ANOVA, $p = 0.027$. Error bars indicate S.E. M. across 8 subjects. **b** Position of the first (in blue) and second fixations (in yellow) over the trustworthy face (left, green ROI) and untrustworthy faces (right, red ROI) for monkey 1. Faces are selected from the Todorov's Database of trustworthy faces[46,65]

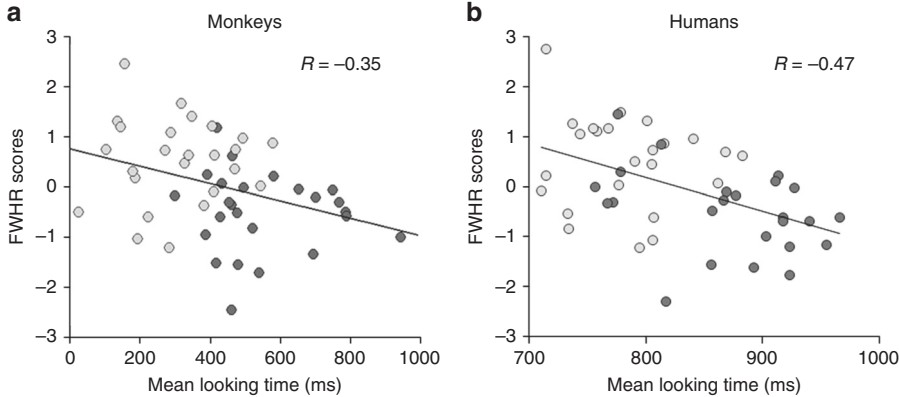

**Fig. 4** Monkeys and humans 'correlation between mean looking time and FWHR. Mean looking time in milliseconds (ms) of (**a**) monkeys ($r(48) = -0.35$, $p < 0.05$, Pearson correlation) and (**b**) humans ($r(48) = -0.47$, $p < 0.001$) are negatively correlated to the facial width height ratio (FWHR) scores, i.e., longer and narrower faces were looked longer by both species. Light grey points correspond to trustworthy faces and dark grey points to untrustworthy ones

$p < 0.001$) (Fig. 4). Thus, long and narrow faces (i.e., lower FWHR) were watched longer by both species.

As FWHR has also been shown to have a link to dominance in monkeys[13,53], we designed an additional experiment (Supplementary Note 3) where we tested the animals' perception of dominant/submissive faces, using the same procedure followed in the trust experiment. The same monkeys ($N = 8$) performed the preferential looking paradigm on the dominance social dimension. Monkeys looked longer at submissive than dominant human faces ($574.51 \pm 193.88$ ms and $447.01 \pm 209.33$ ms, respectively), but the difference between the two types of face was not significant (paired-sample $t$-test, $t(7) = 1.66$; $p = 0.13$).

### Discussion

Our results show a preferential looking to trustworthiness-associated facial cues in monkeys and humans. We further observed that monkeys' visual exploration strategies are differently coordinated when attending trustworthy and untrustworthy faces. When monkeys were looking at trustworthy faces, eye gaze between the first and the second saccade, shifted closer to the eye region, suggesting an approach behavior toward faces bearing trust characteristics. Finally, we found a significant correlation between facial width-to-height ratio and looking time in both species.

This is the first comparative study reporting spontaneous sensitivity to trustworthiness-associated facial cues and related visual attention strategies, in both macaque and human species.

Over the course of visual exploration this preference emerged quickly after stimulus onset in both monkeys and humans: monkeys settled on the preferred (trusting) face after 510 ms while humans at 200 ms.

Furthermore, the analysis on monkeys' gaze patterns showed that their visual exploration strategies are differently coordinated when attending trustworthy and untrustworthy faces: the monkeys' first fixation was longer and followed by a second fixation spatially higher on the trustworthy face but not on the untrustworthy one. The paradigm used in our study does not allow concluding in a definitive way whether monkeys preferred to approach the trustworthy faces or to avoid the untrustworthy ones. However, given that direct eye contact in primates often serves to assert dominance status, the fact that monkeys took a closer look at the trustworthy face's eye region after a prolonged first fixation, seems to suggest that they might have considered these faces more approachable and positive in line with humans' spontaneous preference for this kind of stimuli.

What are the features of trustworthy human faces that attract monkeys' attention? Social traits inferences are constructed from multiple sources of information. In humans, in addition to the physical facial features contributing to perceived femininity and emotional valence[9], the facial structure seems an additional dimension contributing to the perception of trustworthiness. Particularly, faces with lower FWHR are more likely to be judged as trustworthy[12]. In our study we showed that total viewing time on a given human face was negatively correlated to the face FWHR in both monkeys and humans. Thus, long and narrow faces (small FWHR) were looked longer in both species. Our results are in agreement with the findings reported by Stirrat and Perrett (2010) but using this time an implicit measure of visual preference rather than explicit judgements.

In monkeys, FWHR is related to dominance status[13,53]. In macaque societies, staring at the dominant individual is considered a challenge and may lead to harmful consequences, whereas looking at the non-dominant - and non-threatening - individual is clearly a safer avenue[54]. Assuming monkeys generalize this morphological characteristic of their own species (i.e. assuming that they exhibit a "simiomorphic" bias), part of their preference for trustworthy human faces may result from high FWHR signaling caution and low FWHR approachability. In other word, FWHR might be a reliable cue kept through evolution to implicitly communicate trustworthiness from faces.

Although this might seem to be a parsimonious interpretation, avoidance of faces with low FWHR is not sufficient to explain the differential attraction for trustworthy and non-trustworthy faces. Our correlation results show that FWHR accounts for 12.2 and 21.2% of the variance in looking in monkeys and humans, respectively. Furthermore, although monkeys tend to look longer at non-dominant than dominant human faces, this difference was not significant. Thus, other sort of cues must contribute to the monkeys' preference for trustworthy faces. Some of these are likely to be shared with facial cues for femininity, happiness or attractiveness, as evaluations of these social traits correlate strongly with trustworthiness judgements made by human subjects. For instance, macaque monkeys discriminate male vs. female human faces[55]. We could speculate that cues to femininity combined with FWHR and others yet to be determined physical features also signal approachability, and, generates the global impression of trustworthiness. Finally, human faces with small FWHR may resemble most macaque faces, which would explain their preference.

Our findings support the hypothesis that monkeys who are able to infer some aspects of the personality of their conspecifics might recycle this measurable species-typical facial trait, for making similar inferences about human faces. In humans, in addition to the objective FWHR feature a mechanism of emotion overgeneralization has also been considered important for trustworthiness judgements. According to the emotion over-generalization hypothesis, resemblance of neutral faces to emotional expressions is perceived as indicating the trait attributes associated with these emotions[14,17,56]. An emerging explanation for monkeys' preference for trustworthy human faces is that expertise with human faces enables them to detect gender and, possibly, the face general emotional valence, thus facilitating the perception of trustworthy faces as more positive and approachable than untrustworthy ones.

The exploration of correlation between age and the preference toward trustworthy-associated facial cues suggest that experience may also be responsible for the expression of this bias, though, as a note of caution, this needs to be confirmed with a larger group directly examining the effect of age.

Our results are also in line with a number of studies showing that macaques' social abilities extend beyond their own species by encompassing the ability to understand interactive behaviors within the human repertoire. Monkeys can observe and interpret human social cooperation, by preferring to interact with individuals who demonstrate reciprocity with peers[39]. They also spend more time looking at humans that imitate their gestures (lip-smacking) but not at those who previously just stared at them[57]. If monkeys can distinguish humans who reciprocate from those who don't, this suggests that they are attentive to visual social cues emitted by our species[58–60]. Such comprehension of human social behavior might also be the basis of monkeys' ability to form human-like "first impression" of human faces as our results seems to indicate. Darwin proposed that facial displays of emotions serve to predict an individual's current intentions[61] and there is some evidence in nonhuman primates that they can use faces to inform behaviour[62,63]. Inference of social trait is a cognitive mechanism that allows prediction of others' future behavior[17]. Invariant and morphological aspects of the face have a fundamental role in making these inferences. Considering the present findings, it is reasonable to assume that the implicit visual preference that monkeys and humans displayed is made possible thanks to a strong predisposition to use not only overt emotional cues but also stable face characteristics announcing covert social attitudes.

Using a set of standardized human faces, we have shown that macaques respond to facial human features linked to trust. Does this mean that monkeys are responding to the social trait of trust as we assume humans do? Monkeys' preference for looking at trustworthy more than untrustworthy faces, and not the other way around, support this interpretation. Although the meaning of trustworthiness may be different between the two species as the presence of language in humans may further shapes the impression of trust and semantically enriches the concept in a categorical manner, here the behavioral response do not differ between the two species. We might assume that at an implicit level for both species a trustworthy face enhances approach behavior.

Because we used models of parametrized human faces, it is unknown whether similar preferences would have been recorded if macaques were shown virtual parametrized monkeys faces. Given that interspecies abilities in the social domain are more difficult to prove, in the light of the present result, it is reasonable to assume that if macaques show spontaneous preference for human faces conveying trustworthiness they might also be able to do so for faces of conspecifics.

To our knowledge, data on what might constitute, for a monkey, a trustworthy or an untrustworthy conspecific's face, or whether facial features that convey this social trait in humans are present in monkey faces' and ecologically meaningful to a monkey observer, is not available yet. Investigating further whether monkeys exhibit first impression effects for trust in conspecifics' faces would be interesting, but quite challenging. Monkey faces generated with human-defined transformation rules may have very poor ecological validity. Yet, an interesting future study would be to manipulate fWHR of monkey face pictures and assess looking preference of monkey observers. However, as fWHR facial metric correlates with different social judgements[64], this approach may not fully capture the trustworthy features of monkeys' faces. A remaining option is to conduct an extensive ethological study aimed at identifying facial characteristics of more or less trustworthy/approachable individuals within a macaque social group, apply empirically derived transformation rules to generate an appropriate standardized set of macaque face stimuli and assess monkeys' viewing preferences for such stimuli.

Physiognomy is the ancient art of connecting facial features with the underlying character. It is unlikely and unexpected that judgements on social traits based on facial features are always accurate; however, there might be a reason why evolution is keeping the mechanisms necessary to be sensitive to trustworthiness facial features. Detecting fast who can be approached and who should be avoided may constitute a basic reflex-like mechanism intrinsically tied with all primates' social survival, and further modulated by learning and experience acquired while interacting with others.

## Methods

**Subjects**. Monkeys: eight adult monkeys (Macaca Mulatta, one female and four males 4–17 years old, and Macaca fascicularis, three males, 6 years old) have been tested. All animals were born in outdoor enclosures and were then socially housed indoor, so they have been exposed to both conspecifics and humans. All experimental procedures were in conformity with current guidelines and regulations on the care and use of laboratory animals (European Community Council Directive No. 86–609) and were conducted in authorized facilities (Department of Veterinary Services, Health & Protection of Animals, permit number 69 029 0401). The specific research protocol was examined by local and national ethics board, which approved the methods (authorization No. 2015061213048343). Humans participants: Twenty human participants, (10 women, M age = 26.3 years, SD = 7.1), with normal or corrected vision, gave written informed consent to participate in the experiment. Subjects were blind about the hypotheses of the study but they knew they were participating in a first impression study. Human experiments were sponsored by CNRS, approved by the ethical committee Sud-Est IV, Lyon and performed in accordance with the Declaration of Helsinki. A initial experiment was performed with fifty-four healthy subjects, (27 women, 26.9 ± 5.9 years) using a 5 s exploration duration (Supplementary Note 2)

**Stimuli**. The stimuli were 48 computer-generated male faces created with the FaceGen 3.1 software development kit (Singular Inversions, Toronto, Canada) selected from the Todorov's well-controlled quantitatively validated stimulus repertoire of faces. All faces were bald and Caucasian. The Trustworthiness database is composed of facial identities varying on seven levels of trustworthiness[46,65]. Todorov et al' work showed that human explicit judgements of trustworthiness match with the model's prediction[46] For the current study we selected the most ( + 3 SD, $N = 24$) and the least (−3SD, $N = 24$) trustworthy faces. On each trial a couple of the same identity differing only for their level of trustworthiness-associated features was presented.

**Task procedure**. Monkeys: to assess preference formation, we used a preferential looking paradigm. For monkeys, pairs of faces were presented in a random order. Each face pair was presented twice to counterbalance for side of presentation. During the experiment, monkeys were seated in a primate chair inside a darkened room with their head restrained. Stimuli were presented on a 15-inch color monitor (1024 × 768 pixels) at a viewing distance of 24 cm. A trial began with the appearance of a single fixation point in the center of the screen. Once the monkey fixated this point, two face stimuli subtending 13° x 21.2° of visual angle (207 × 340 pixels) were displayed and remained on the screen for up to 2 s. The monkey was free to move its eyes over the images and received a juice reward provided its gaze stayed within the boundaries of the video monitor for the entire 2 s period, otherwise the stimuli were extinguished and the trial discarded. Monkeys could

choose to look outside of the face and still receive the reward. The monkey's gaze position was monitored by ISCAN infrared eye tracking system at 200-Hz. Experimental control, stimulus presentation, data sampling and storage was done with REX/VEX software system. Before the experiment, monkeys underwent a 5-point eye position calibration and were trained until they understood the visual exploration task using 8 different pairs of non-face biological and non-biological stimuli. Humans participants: healthy participants ($N = 20$) were instructed to look at the same pairs of faces. We designed a 2 s exploration duration task because preliminary testing showed that monkeys' attention decreased and gaze escaped from the screen with longer stimulus presentation. Stimuli were displayed on a 17-inch computer screen at a resolution of $1280 \times 1024$ pixels using Presentation® software (Version 14.9, www.neurobs.com). The viewing distance from the participant's eyes to the screen on which stimuli subtending 7.8° x 12.5° of visual angle ($377 \times 604$ pixels) were displayed was 73 cm. Humans' eye positions were recorded using an infrared video-based tracker (Tobii 1750) at a 60-Hz sampling rate and Clearview 2.7.0 allowed online recording of eye-gaze data. The two systems were synchronized using the Tobii extension for Presentation. In a second session the same pairs of faces were presented and humans were asked to explicitly select the most trustworthy faces. Eye positions were recorded until the response. During all sessions the experimenter monitored on-line the position of the subject's eye gaze that was projected on a second screen in the same room but placed far from the location of the participants. Prior to the experiment, humans underwent a 5 point-calibration task. The final experimental set comprised 48 trials.

**Pre-processing and data analysis**. ClearView fixation filter was used to filter the data for humans (with a visual angle of 1° and duration of 100 ms). An in-house Matlab script was used to pre-process and filter monkeys' eye-tracking data. First, eye velocity for each location was computed as the angular distance traversed by the eye within a 5 ms moving window. Next, for each trial, a velocity threshold was set at three times the median during the 2 s window. Data points that exceeded this threshold were considered as saccades. Fixation times were considered as the interval between two saccades with a minimum duration of at least 100 ms, and fixation locations were defined as the eye position at the central fixation time point. In order to quantify allocation of attention to faces, regions of interest (ROI) delimiting each face were defined manually. The mean looking time was calculated as the average of the total time spent within each ROI during a trial. Only trials with at least one fixation at one of the two faces were included in the dataset. For the main statistical analysis, mean looking times on each face were calculated for each participant and for each trial.

**Visual exploration density**. To estimate whether monkeys and humans preferentially gazed within the faces' area compared to the rest of the screen (Fig. 1), we submitted participants' exploration density over the screen to three Gaussian-mixture models, based on a selection procedure applying the Akaike information criterion (AIC). The first model, a "central bias Gaussian model", represents density of exploration as a Gaussian function centered on the screen, and therefore this is the model that best fits with the central bias behavior usually shown by humans while looking at scenes. The second, a "two faces Gaussian model", density of exploration is exemplified by two Gaussian functions, each centered on one of the regions of interest (Left or Right face). Finally, the third, a "two faces and central bias Gaussian model", estimates density exploration behavior by combining the two previous models. We considered as the best model the one that reports the lowest AIC values because of the quality of the fit and the complexity of the model. In monkeys, (Fig. 1), the best model was the "two faces and central bias Gaussian model" (AIC = $6.44 \cdot 10^5$) compared to the "central bias Gaussian model" (AIC = $6.47 \cdot 10^5$). In humans, a similar result was found ("two faces and central bias Gaussian model" with AIC = $4.43 \cdot 10^5$ while "central bias Gaussian model" reached a higher AIC = $4.86 \cdot 10^5$.

**Temporal dynamics and statistical analysis**. In order to identify the time windows showing significant differences between trustworthy and untrustworthy faces, a large-scale multiple testing procedures was designed. First, for each subject, the probability of fixation for each face type was computed at each time point. Second, each individual curve was slightly smoothed with a Savitzky-Golay filter of 51 ms length and polynomial order 2. Then, statistical differences between the averaged probabilities of fixation in trustworthy and untrustworthy ROIs were tested using $t$-tests at each time point (from 200 ms to 2 s post-stimulus onset). Finally, continuous periods were a fixation bias can be observed on one of the face stimuli were identified using a cluster-based permutation test (alpha cluster = 0.05; number of permutation = 100)[66].

**Spatial distribution of fixations**. To provide information on the spatial distribution of the fixations, the barycenter of fixations and a heat map representation were calculated for each face at the subject level. Heat maps were calculated using Gaussian kernel density mapping of the fixations, weighted by the fixations' duration[67]. Then, at the group-level, individual heat-maps were normalized and averaged to visualize the spatial distribution of the fixations of the studied population.

**Facial width-to-height ratio (FWHR)**. To obtain a score for the facial width-to-height ratio, two independent raters measured the distance between the lip and brow (upper facial height) and the left and right zygion (bizygomatic width) of each face from the whole database. FWHR was calculated as width divided by height[52]. Inter-rater reliability was high for all measures (all $rs > .79$, all $ps < 0.001$).

**Code availability**. Matlab code used in this project for Data analyses are available upon request to the corresponding author.

## Data availability
Raw data are available upon request to the corresponding author.

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

## Acknowledgements

This research was funded by CNRS, ANR and by LABEX CORTEX (ANR-11-LABX-0042) University of Lyon, within the program "Investissements d'Avenir" (ANR-11-IDEX-0007) to AS and JRD. MC is supported by a French Minister of Research fellowship. We thank Andres Posada for help during eye movement analyses in human subjects and Suliann Ben Hamed for allowing testing one monkey.

## Author contributions

A.S. proposed the study concept. All the authors designed the methods. Data collection was performed by M.C. and A.G. (human participants) and by E.B. (monkeys). Data analysis was conducted by M.C., E.B., A.G. and G.L. under the supervision of A.S. and JRD. M.C., A.G., A.S. and J.R.D. wrote the paper. All authors discussed the findings and approved the final version of the manuscript for submission.

## Additional information

**Competing interests:** The authors declare no competing interests.

