## [Peer Review File · Nature Communications]

Reviewers' comments:

Reviewer #1 (Remarks to the Author):

Costa and colleagues investigate whether monkeys show a preference for viewing faces that is dependent on their perceived trustworthiness (based on prior studies that have investigated this characteristic in humans). They find that both monkeys and humans exhibit a preference to look at more trustworthy faces. Further, monkeys showed an upward eye movement to the eye region when viewing trustworthy faces, which the authors interpret as an approach behavior. Finally, they report correlations between the width-to-height ratio of the faces and the looking time in both monkeys and humans.

This is a very interesting comparative study that suggests a common basis for evaluation of faces in monkeys and humans. In general, the experiment is well-conducted, with data from a reasonably large number of monkeys ($n = 8$) and the manuscript well-written. But I think there are some specific concerns the authors need to address to strengthen their findings.

1) It's a little concerning that the stimulus duration was not matched between species (humans, 5 s; monkeys 2 s) and it can't be ruled out that some of the differences between the species results from this change. I would strongly recommend the authors collect some additional human data with the same presentation duration as the monkeys and see if duration effects any of the patterns observed.

2) Chance level was computed as the ratio between the number of pixels on the ROI and the total number of pixels on the screen. I think this may be a misleading estimate since there's a well-known central bias for fixations (i.e. people do not make fixations near the edges of the screen/stimulus). I'm not sure of a good solution to this concern - could the authors use eye movement data from scene stimuli to estimate what the effective potential fixation area of the screen is and use this value as the denominator in their ratio?

3) The introduction and discussion are very long and quite meandering. The manuscript would benefit from a much tighter introduction that references the essential background only, and a discussion that minimizes the currently extensive speculation and stays closer to the data.

4) The authors cite work by Sugita suggesting that monkeys deprived from seeing faces still show a preference for faces over other objects. The authors might also want to consider the work from Marge Livingstone's lab (e.g. Arcaro et al, 2017) that suggests a slightly different picture.

5) I highly recommend the authors show the data points/lines for all 8 monkeys individually in some of their figures (e.g. Figure 1A, Figure 2).

6) Do humans show any specific bias similar to the monkeys' upward fixations for trustworthy faces on the second saccade?

7) In Figure 2 it would be helpful to provide some indication of how the y coordinate values correspond to a face stimulus. How high was the face? Is a value of 50 a large difference in location? Perhaps the authors could show a face to the left of the y-axis to provide some measure of scale relative to the stimulus.

8) In the second sentence of the Discussion the authors highlight the results of the mean duration of the first fixation analysis. But this analysis was only conducted for monkeys that showed a preference based on trustworthiness (or a trend) and I think the description is potentially misleading without the context. Moreover, I'm also not entirely convinced that limiting the analysis in this way is appropriate.

9) Page 13: "Since we presented two faces simultaneously without giving explicit instructions to participants, it is not surprising that human preference for trustworthy faces emerged later in time". I don't see how this follows – the authors need to unpack and justify this statement.

10) Page 19, Stimuli: "For the current study we selected from the 24 identities the two most extreme versions (-3SD and +3SD)". This sentence is unclear. Presumably, the authors mean they selected the 24 most trustworthy and the 24 least trustworthy?

Reviewer #2 (Remarks to the Author):

This paper reports a novel and quite surprising finding: macaque monkeys show a preference for trustworthy-looking human faces. It is well established that people hold specific stereotypes about trustworthy appearance and act on these stereotypes. The faces used by the authors were generated by a model that visualizes these stereotypes. The finding that macaques show a preference similar to humans is surprising, because the facial cues are fairly subtle (the fWHR is a very crude cue that happened to correlate with judgments of trustworthiness; see below).

The authors should take a look and cite a highly relevant paper by Jessen & Grossman (2016). *Journal of Cognitive Neuroscience*, 28, 1728-1736. These authors used the same preferential looking paradigm and the same stimuli to study the behavior of 7-month old human infants. The findings are remarkably similar to the present findings: infants prefer to look at trustworthy than at untrustworthy looking faces, but show no such discrimination for dominant vs. submissive looking faces. These findings provide additional credence to the present findings.

Yet it is easier to explain the infants than the macaque's findings. By 7-months of age, infants can discriminate positive and negative expressions of emotions and are most likely to have a woman as a primary caregiver: cues that are correlated with perceptions of trustworthiness.

It is harder to make the same case for monkeys unless one invokes some sort of social learning from observing humans. The authors seem to invoke a nativist explanation but then it is important to show that the same configurations of features in monkey and human faces trigger similar perceptions of approach/avoidance. I would also suggest looking at the recent work of Margaret Livingston from Harvard Medical School. Her group raised monkeys without visual exposure to faces (similar to Sugita) and then used fMRI to search for face selective regions. The findings suggest that such regions are not formed unless monkeys are exposed to faces. This suggests that one needs an extensive learning and exposure to faces to develop the proper specialization and sensitivity to minor differences between different faces.

This learning interpretation doesn't take anything away from the originality of the authors' findings and needs to be discussed as a plausible mechanism. Looking at Suppl Table 1, the youngest monkey was 4 years old and the oldest 17. Unfortunately, the sample is too small to estimate the relationship between age as a proxy for experience with humans and bias to look at trustworthy faces (nevertheless, a quick analysis shows a correlation of .36; the scatter plot is reasonable with one outlier).

The authors invoke the FWHR as a possible universal feature: "FWHR might be an objective feature kept through evolution to implicitly detect trustworthiness from faces." However, as they point out, this feature accounts for very little of the variance of trustworthiness judgments. Initially, this feature was interesting to psychologists and evolutionary biologists because of its possible sexual dimorphism, but recent meta-analyses show that its correlation with gender is minuscule relative to correlations with body/height and upper body strength. In humans, the measure is also correlated with body-mass index and this correlation can explain much of the observed effects in the literature.

The overgeneralization mechanism can serve as a potential explanation of the findings if monkeys are sensitive to emotional expressions in humans; presumably, more caring humans would have more positive expressions when handling the monkeys and this might be consistent across situations. This seems to be consistent with the fixation patterns of the monkeys (Fig. 1C), patterns clustered around the nose/mouth.

In sum, sound research with an interesting and provocative finding in need of a good explanation.

Reviewer #3 (Remarks to the Author) (See also attached PDF)

General comments:

The manuscript reports findings from a comprehensive comparative study investigating implicit preference for human faces characterised by anatomical features associated with trustworthiness. The findings show that macaques show a preference for trustworthy human faces, but with a slightly different pattern of attention to humans.

The paper is well written, clear and complete. The design is appropriate, and the results seem strong. I have to admit that I am not very familiar with the analysis of eye tracking data, but they seem sound. I like the direct comparison between macaques and humans using the same methods. One addition that could be envisaged is some analyses of pupil dilation, which would the authors to have stronger interpretation of the attention pattern. If pupil dilation, and therefore arousal, is higher when looking at untrustworthy faces this could suggest that macaques avoided them. If pupil dilation was higher for trustworthy faces this would strengthen the approach interpretation.

Overall, the interpretations of the results are suitable, but I would like to see a bit more caution at times and more consistency in the use of some key terms. The discussion is interesting but features quite a few repetitions, which makes it quite lengthy. Perhaps this could be reorganised in a more concise way.

I believe the contribution of this paper is worthy of publication in Nature Communications, provided that the authors make a few changes. Given the comparative nature of the study and the methods used, I think this work will be of interest to a broad readership.

Detailed comments

There were no line numbers so see the attached file for detailed comments.

Implicit preference for human trustworthy faces in macaque monkeys

Manuela Costa¹, Elodie Barat¹, Alice Gomez¹, Guillaume Lio¹, Jean-René Duhamel^{1*}, Angela Sirigu^{1*}

1 Institut des Sciences Cognitives Marc Jeannerod, CNRS, UCBL, Lyon 1, Bron, France

* “These authors contributed equally to this work”

Correspondence to: sirigu@isc.cnrs.fr

Abstract

In numerous species, trust is a basic prerequisite of group living. As it does not come without a risk, the ability to select trustworthy partners is an essential survival skill. Research in social psychology has demonstrated that human judgments of trustworthiness are based on subtle processing of specific perceptual features. However, it is not known if this ability is shared among primates or is a specific human function. Here we report that macaque monkeys (*Macaca Mulatta* and *Macaca Fascicularis*), like humans, display a preferential attention to trustworthiness-associated facial cues portrayed in computer-generated human faces. They looked significantly longer at faces categorized *a priori* as trustworthy compared to untrustworthy. In addition, spatial sequential analysis of monkeys' initial saccades revealed an upward shift with attention moving to the eye region for trustworthy faces while no change was observed for the untrustworthy ones. Finally, we found significant correlations between facial width-to-height ratio – a morphometric feature that predicts trustworthiness' judgments in humans –and looking time in both species. These findings suggest the presence of common mechanisms among primates for first impression of trustworthiness.

Keywords: face processing, human and non-human primate, trust

Introduction

Trust is a fundamental psychological dimension, influencing people's decisions in social interactions such as cooperation ^{1,2}, voting intentions ³, economic decision-making ^{2,4}. Trusting is taking the risk of putting one's own fate in someone else's hands, hence the importance of trustworthiness assessment to minimize this risk.

Surprisingly, research in social psychology has demonstrated that rather than being based solely on rational criteria (reputation, prior social interactions), judgments of trustworthiness in humans are robustly related to specific perceptual features ⁵⁻⁷. For instance, different shapes of eyebrows, cheekbones and chin can trigger different perceptions of trustworthiness from faces. These facial features automatically capture observers' attention and lead to trustworthiness judgments after an exposition to single face as brief as 33ms, the so-called first impression effect ⁸. This effect is based on detection of facial cues and also on holistic processing of a face's appearance.

It has been shown that the facial width-to-height ratio (FWHR) ⁹, a morphometric measure that relies on face structure, predicts explicit judgments of trustworthiness ¹⁰. For instance, faces with small FWHR are more likely to be judged as trustworthy ¹⁰. Similar results have been found in in Capuchin Monkeys where a study has demonstrated a link between FWHR and social dominance ¹¹. This raises the possibility that species-typical facial traits, which are objectively measurable, are used by monkeys to infer the "personality" of conspecifics, and to push even further this idea, the same processes might be recycled for making similar inferences about human faces. In humans, other facial attributes also predict trustworthiness judgments, including the resemblance of faces to femininity ¹²; facial maturity ¹³, physical similarity to the self ¹⁴ or proximity with positive emotional expressions. Todorov and colleagues proposed the overgeneralization of emotion as the main mechanism underlying perception of trustworthiness ¹⁵. According to this hypothesis, neutral faces that do not display any emotional expression are nevertheless perceived as expressing behavioral tendencies associated with the emotion the face

resemble most ¹⁶⁻¹⁸. For instance, trustworthy and untrustworthy faces are perceived as resembling happy and angry faces, respectively ^{12,15,18,19}.

The importance of facial information in social interactions and survival has been revealed in humans and monkeys, suggesting that homologous neural and behavioral mechanisms might exist in different primate species ²⁰⁻²². Because the capacity to perform judgments of trustworthiness strongly involves processing of facial cues, and given the adaptive value of this skill in cooperative societies, one may wonder whether such ability has an evolutionary origin. In this study, we asked if non-human primates, just like humans, are responsive to trustworthiness-associated facial cues. We addressed this question by recording monkeys' eye movements using a preferential looking paradigm.

The rationale for using this approach relies on the known fact that non-human primates are highly sensitive to, and make use of facial cues during social interaction.

Along this idea, developmental studies have shown that infant rhesus monkeys exhibit both innate and early experience-dependent preferences for both human and non-human primate faces ^{23,24}. Non-human primates are indeed strongly attracted by faces ^{23,25,26}. This behavior is innate as newborns rhesus macaques deprived from seeing their mother's or caregivers faces still show preference for faces compared to objects ²³. This preference recalls a similar one observed in human newborns who spend more time looking at faces compared to objects ^{27,28} suggesting that preference for faces is an innate skill which evolves during early development ²⁹⁻³¹.

Other studies have also shown that adult macaques and humans are sensitive to specific facial features. Rhesus macaques are sensitive to face identity ^{32,33}, they show preferences for conspecifics' faces compared to other species ³⁴⁻³⁶, although being also interested by human faces. Furthermore, they are sensitive to familiarity ³⁷, and they easily detect facial expressions ³⁸.

Despite evidences pointing to macaques' early abilities in processing distinct facial components, one may still wonder why this species should be sensitive to human facial traits of trustworthiness. Additional arguments can be advocated in favor of this hypothesis. First, monkeys

bred and raised in captive environments have experience with humans from their earliest age and develop considerable expertise about our physiognomy. For instance, we have shown that macaques recognize the identity of familiar humans in both face pictures and voice samples, and spontaneously match known faces to the corresponding vocal signatures³⁹. Thus, it is not unreasonable to assume that these animals learn about human approachability and trustworthiness and that they can associate these behavioural traits with observable human characteristics, including facial features.

Second, just like human newborns⁴⁰ baby monkeys imitate human adult facial movements such as tongue protrusion or lip-smacking thus showing early abilities in recognizing and reproducing human facial features⁴¹.

Third, they are sensitive to observed human interactions. They show avoidance of humans who are not helpful and do not reciprocate in social exchanges^{42,43} and they approach and look more at humans who are imitating them⁴⁴. Eye tracking studies have further shown that non-human primates look more frequently and longer at positive valence stimuli signaling approach behavior and less frequently and shorter at negative stimuli associated with withdrawal behavior⁴⁵.

Finally, the visual system of monkeys and humans show strong homologies and, notably, the same temporal lobe's functional organization into multiple, hierarchically organized face patches^{21,46}. The idea of a shared neural mechanisms for processing conspecific faces and human faces within these circuits is confirmed by a large number of single unit recording studies showing that macaque specialized areas contain intermingled monkey-selective and human-selective face neurons^{35,47,48}

In the light of these evidences we reasoned that monkeys might be able to process facial features and discriminate between trustworthy and untrustworthy human faces by using first impression mechanisms as hypothesized for humans⁴⁹. In this study, macaque monkeys looked at pair of faces differing for their level of **trustworthiness**. We hypothesized that attention towards one of the two faces could be a sign of detection of the features that differently characterize the pair of

faces. Because monkeys were not rewarded to specifically look at faces we assumed that significantly longer looking time towards trustworthy faces may be interpreted as a preference towards those stimuli signaling an approach behavior.

We showed macaque monkeys ($N=8$) pairs of parameterized human faces drawn from Todorov et al's image database ⁴⁹, each displaying a most (+3SD from the baseline) and a least (-3SD from the baseline) trustworthy version of the same facial identity. These computer-generated faces only vary on the facial features that predict judgments of trustworthiness. Human faces with high inner eyebrows, pronounced cheekbones, wide chins and shallow nose sellion are perceived as more trustworthy than faces with low inner eyebrows, shallow cheekbones, thin chins and deep nose sellion ¹⁹. We presented the two extreme variants of the same facial identity in each trial to ensure that monkey's preference towards one face or another could depend from the only difference between the two stimuli on trustworthiness-associated facial cues. To ensure spontaneous preferences, monkeys were not rewarded to look at faces. They freely moved their eyes about and were periodically given juice rewards to maintain gaze within the limits of the computer screen surface where the images were displayed (Materials and methods). To establish across-species comparisons, and assess if humans would also spontaneously allocate gaze toward trustworthy faces, we assessed the performance of human subjects ($N=54$) following the same procedure as in monkeys.

Results

Monkeys' and humans' visual preferences

In order to quantify gaze allocation, regions of interest (ROIs) encompassing the trustworthy and untrustworthy faces were defined. Ocular fixations within and outside these ROIs were recorded during each trial (Materials and methods). The mean looking time was calculated as the average of the total time spent within trustworthy and untrustworthy faces for all stimulus pairs presented.

The first analysis, as expected, revealed that monkeys were attracted to both faces, spending more time on these stimuli than predicted by random exploration of the video monitor (chance level=160ms; trustworthy: mean±s.d = 512.89 ± 223.87ms, $t_{(7)} = 4.45$, $P < 0.01$; untrustworthy: mean±s.d = 292.60 ± 162.67ms, $t_{(7)} = 2.30$; $P = 0.054$) (Fig.1). Furthermore, monkeys discriminated between the two stimuli presented and spent significantly more time looking at trustworthy than untrustworthy faces (paired sample t-test; $t_{(7)} = 3.29$; $P < 0.05$, $\eta^2 = 0.60$) (Fig.1a). This visual preference was remarkably consistent across animals, as 3 monkeys (O, Y, T) showed a highly significant bias, 1 (E) a nearly significant bias, 4 a non-significant positive bias toward the trustworthy faces. Importantly, none of the monkeys showed the opposite trend (Supplementary Table 1).

Humans followed the same pattern, spending most of the time looking at the faces (chance level=853ms; trustworthy: mean±s.d =2311.96 ± 228.72ms, $t_{(53)} = 37.87$, $P < 0.01$; untrustworthy: mean±s.d = 2177.11 ± 208.75ms, $t_{(53)} = 37.17$, $P < 0.01$), showing a significant bias in favor of the trustworthy face category (paired sample t-test; $t_{(53)} = 2.96$, $P < 0.005$, $\eta^2 = 0.14$) (Fig.1d).

Fixations frequency analysis revealed the same significant preference for both species. Monkeys performed more fixations over trustworthy than untrustworthy faces (trustworthy: mean±s.d = 1.44 ± 0.48; untrustworthy: mean±s.d = 0.92 ± 0.42; paired sample t-test; $t_{(7)} = 3.24$; CI 0.14 - 0.89; $P = 0.0142$). Humans exhibited the same pattern, showing a significant increase in number of fixations in favor of the trustworthy faces (mean±s.d = 6.66 ± 1.60; untrustworthy: mean±s.d = 6.33 ± 1.48; paired sample t-test; $t_{(53)} = 2.44$; CI 0.058 - 0.59; $P = 0.0179$).

In order to determine whether longer looking times were actually related to the perception of trustworthiness, in a separate session, humans were explicitly asked to select the most trustworthy face while again their eye movements were recorded. We found that the difference in mean looking times (trustworthy-untrustworthy) was significantly correlated between the first (implicit) and the second (explicit) condition ($r = 0.30$, $P < 0.05$) (Supplementary Fig. 1). This shows that humans'

spontaneous looking times toward trustworthy faces in the non-instructed viewing task is a reliable implicit marker of trustworthiness detection.

Hence, our results reveal that both macaque monkeys and humans detected and preferred to look at human faces displaying trustworthiness-associated facial cues.

Because of this common preference across species, we explored whether monkeys and humans used similar eye gaze strategies with a focus on temporal dynamics and spatial distribution of fixations. A cluster-based permutation test (Materials and methods) showed that, in monkeys, preference for the trustworthy faces occurred from 510ms to 1485ms after image onset ($P < 0.05$ corrected for multiple comparison) (Fig. 1b), while humans' preference occurred in two stages and later in time, with a first short-lived preference emerging at 1760ms and a more stable one at 3640ms ($P < 0.01$) (Fig. 1e). To provide information on the spatial distribution of visual exploration (Materials and methods), heat maps and barycenter of eye fixations were generated (Fig. 1c-f). Overall, monkeys preferentially allocated their attention in the region surrounding the nose (Fig. 1c), while humans eye gazed mostly around the eye region (Fig. 1f).

Fig. 1. Looking preference for trustworthy vs. untrustworthy faces by rhesus macaques and human subjects. MONKEYS (N=8): (A) Mean looking time in milliseconds (ms) for the most trustworthy (+3SD of the neutral face) and the least trustworthy (-3SD of the neutral face) versions of the same facial identities. The error bars denote standard error of the mean. $*P < 0.05$. Monkeys looked significantly longer at the two faces than predicted by chance and looked more at trustworthy than untrustworthy faces (chance level represented with dotted line was 160ms for each face region of interest). **(B)** Time course of looking preference. Mean viewing time on each facial prototype plotted each 15ms. A cluster-based permutation test showed that preference for the trustworthy faces (green line) was significant between 510ms and 1485ms ($P < 0.05$ corrected for multiple comparison). **C.** Gaze heat maps for trustworthy and untrustworthy faces averaged across subjects (trustworthy face on the left by convention, facial prototype spatial location was counterbalanced within and between subjects). Yellow dots show fixation centers of gravity for each subject. **HUMANS (N=54) (D-E-F)** Plots show (D) significantly longer mean looking times at trustworthy than untrustworthy faces and (E) later onset of preference for trustworthy faces (1760ms to 2615ms and again from 3640ms to the end of the trial). Note that the average barycenter of fixation was located in the region surrounding the nose in monkeys whereas it is around the eye region in humans (C, F).

Spatial dynamic of eye fixations and duration of fixations

We hypothesized that monkeys may detect human trustworthiness early after face stimuli onset. We explored monkeys' individual scanning patterns on each trial and for each stimulus type (Trustworthy/Untrustworthy). We computed the mean location (x and y coordinates) and duration of fixations for the first two fixations (no monkeys performed more than two fixations for each face). We reasoned that if monkeys detected quickly the trustworthy face, their visual attention should move towards socially salient regions and therefore spatial coordinates between the first and the second fixation were expected to change. In other words, an approach behavior would be manifested by an upward shift between the first and the second fixation thus bringing the gaze close to the eye region while avoidance would keep the gaze far from the eyes area. We compared across stimulus type (Trustworthy/ Untrustworthy) the mean y fixation coordinate (weighted by the fixation duration) of the first two fixations within the face area.

The ANOVA performed on the weighted y coordinate with the fixations (first or second) and the face type (Trustworthy vs Untrustworthy) as within subject factor revealed no main effects of face type $F_{(1,6)} = 2.32$, $P = 0.18$ and no main effect of fixation order: $F_{(1,6)} = 1.34$; $P = 0.29$. However, we

found a significant interaction effect of stimulus type X fixation order: $F_{(1,6)} = 8.38$; $MSE = 1014.3$; $\eta^2 = 0.58$; $P = 0.027$ (Fig. 2). Post-hoc analysis showed that the location of the first fixation was identical on both faces type (trustworthy y coordinate: $\text{mean} \pm \text{s.d} = 355.7 \pm 13.71$; untrustworthy y coordinate: $\text{mean} \pm \text{s.d} = 357.3 \pm 21.12$) whereas, for the trustworthy face, the second fixation was located closer to the eye region compared to the second fixation on the untrustworthy face which landed close to the mouth region (trustworthy y coordinate $\text{mean} \pm \text{s.d} = 335.1 \pm 10.85$; untrustworthy y coordinate: $\text{mean} \pm \text{s.d} = 360.8 \pm 30.20$; $P = 0.02$, *post hoc t-tests*).

Fig. 2. Fixations sequence analysis. Graph shows monkeys' Y coordinates of first and second fixations weighted by duration (first in blue, second in yellow) and type of face (trustworthy in green, untrustworthy in red); error bars show standard deviations. Fixations closer to the eye region are closer to $Y=300$. The location of the first fixation on the face is not different for the trustworthy and untrustworthy face but the location of second fixation is closer to the eye region only for the trustworthy face suggestive of an approach behavior.

To further understand how fast monkeys' preference toward trustworthy faces progressed in time we performed an analysis on fixation duration on those monkeys showing a significant or marginally significant preference for trustworthy faces.

We selected monkey 1, 2, 3, 4, 7. We compared the duration of their first fixation between face stimulus type. We found that mean duration of the first fixation was significantly longer for trustworthy than untrustworthy faces (trustworthy: mean±s.d = 361.6 ± 90.2; untrustworthy: mean±s.d = 317.3 ± 101.9; paired sample t-test; $t_{(4)} = 3.76$; $P = 0.02$). The same result was also obtained in the human population (trustworthy: mean±s.d = 260.2 ± 73.2; untrustworthy: mean±s.d = 256.6 ± 74.6; paired sample t-test; $t_{(48)} = 2.32$; $P = 0.02$).

These findings corroborate our main findings on looking time by showing that monkeys' visual exploration strategies are differently coordinated when attending trustworthy vs untrustworthy faces. Thus, results on both spatial dynamics and duration of fixations provide evidence for an early and automatic approach behavior toward trustworthy faces in macaques.

Correlation between looking time and FWHR

In humans, FWHR – the ratio between upper facial height and the bizygomatic width - is used implicitly to form social judgments from facial appearance such as trustworthiness¹⁰ and dominance^{50,51}, and we recently showed that the upper facial height is a robust predictor of trust⁵². Considering the potential importance of facial width-to-height ratio (FWHR) in human judgments of trustworthiness¹⁰, we tested whether this morphological character might have a specific role in driving the viewing preference for trustworthiness in monkeys and humans. FWHR was calculated using standard landmarks⁵³. To compute FWHR, two independent raters measured the distance between the lip and brow (upper facial height) and the left and right zygion (bizygomatic width) of each face from the entire image database. Inter-rater reliability was high for all measures (all $R_s > 0.79$, all $P_s < 0.001$). In agreement with Stirrat and Perrett's findings, we found that faces that have been judged by humans as trustworthy displayed a lower FWHR compared to the untrustworthy ones (ANOVA; $F_{(1,24)} = 116.97$, $P < 0.05$; trustworthy: mean±s.d = 0.02 ± 0.019; untrustworthy: mean±s.d = 2.15 ± 0.02). The obtained classification of face stimuli was then regressed against monkeys' viewing preferences. Interestingly, total viewing time on a given face was negatively

correlated to its FWHR in both monkeys ($r_{(48)} = -0.35$, $P < 0.05$) and humans ($r_{(48)} = -0.46$, $P < 0.01$) (Fig. 3). Thus, long and narrow faces (i.e., lower FWHR) were watched longer by both species-

Fig. 3. *Monkeys and humans ‘correlation between mean looking time and facial width height ratio (FWHR) score for each face.* Mean looking time of monkeys (left graph, $r_{(48)} = -0.35$, $P < 0.05$) and humans (right graph, $r_{(48)} = -0.46$, $P < 0.001$) are negatively correlated to the FWHR scores, i.e., longer and narrower faces were looked longer by both species. Light grey points correspond to trustworthy faces and dark grey points to untrustworthy ones.

As FWHR has also been shown to have a link to dominance in monkeys^{11,54}, we designed an additional experiment (see supplementary materials) where we tested the animals’ perception of dominant/submissive faces, using the same procedure followed in the trust experiment. The same monkeys ($N = 8$) performed the preferential looking paradigm on the dominance social dimension. Monkeys looked longer at submissive than dominant human faces (574,51ms, $SD = 193,88$ and 447,01 ms, $SD = 209,33$, respectively), but the difference between the two types of face failed to reach significance ($T_{(7)} = 1.66$; $P = 0.13$).

Discussion

Our results show in monkeys and humans a preferential looking to trustworthiness-associated facial cues. The mean duration of the first fixation on trustworthy faces was significantly longer than the first fixation for untrustworthy faces, thus providing evidence for an early and

automatic preference for trustworthy faces in both species. We further observed that monkeys' visual exploration strategies are differently coordinated when attending trustworthy and untrustworthy faces. When monkeys were looking at trustworthy faces, eye gaze between the first and the second saccade, shifted closer to the eye region, suggesting an approach behavior toward faces bearing trust characteristics. Finally, we found a significant correlation between facial width-to-height ratio and looking time in both species.

This is the first comparative study reporting spontaneous sensitivity to trustworthiness-associated facial cues and related visual attention strategies, in both macaques and humans species.

The emergence of this preference over the course of visual exploration differed in monkeys and humans: monkeys settled on the preferred (trusting) face early on (510ms), whereas humans first explored both faces for about 2s before exhibiting a preference for the trustworthy face. In addition to exposure time to the face stimuli (2s and 5s for monkeys and humans, respectively), the different temporal profiles may be related to the fact that monkeys were rewarded only for maintaining gaze within the limits of the screen, not for exploring the two faces, whereas human subjects may have, wittingly or not, construed the task as requiring exploration and comparison of both images. In humans, explicit judgments of trustworthiness can be reliably obtained with exposure to single face stimuli as short as 33ms⁸. Since we presented two faces simultaneously without giving explicit instructions to participants, it is not surprising that human preference for trustworthy faces emerged later in time. Such pattern was reported in a preferential looking study on attractiveness in which gaze was initially distributed between the two human face stimuli and then gradually shifted towards the face that was later judged as more attractive⁵⁵.

Overall, monkeys and humans were significantly attracted to trustworthy faces despite different temporal and spatial strategies. The analysis on duration of fixations further revealed that the appearance of the spontaneous preference for trustworthy faces can be observed in both species as early as following the onset of the second fixation. This finding corroborates our main results providing evidence for an early and spontaneous preference for trustworthy faces.

Furthermore, the analysis on monkeys' gaze patterns showed that their visual exploration strategies are differently coordinated when attending trustworthy and untrustworthy faces: the monkeys' first fixation was longer and followed by a second fixation spatially higher on the trustworthy face but not on the untrustworthy one. The paradigm used in our study does not allow to conclude in a definitive way whether monkeys preferred to approach the trustworthy faces or to avoid the untrustworthy ones. However, given that direct eye contact in primates often serves to assert dominance status, the fact that monkeys took a closer look at the trustworthy face's eye region after a prolonged first fixation seems to suggest that they might have considered these faces more approachable and positive in line with humans' spontaneous preference for this kind of stimuli.

What are the features of trustworthy human faces that attract monkeys' attention? Social traits inferences are constructed from multiple sources of information. In humans, in addition to the physical facial features contributing to perceived femininity and emotional valence¹², the facial structure seems an additional dimension contributing to the perception of trustworthiness. Particularly, faces with lower FWHR are more likely to be judged as trustworthy¹⁰. In our study we showed that total viewing time on a given human face was negatively correlated to the face FWHR in both monkeys and humans. Thus, long and narrow faces (small FWHR) were looked longer in both species. Our results are in agreement with the findings reported by Stirrat and Perrett (2010) but using this time an implicit measure of visual preference rather than explicit judgments.

In monkeys, FWHR is related to dominance status^{11,54}. In macaque societies, staring at the dominant individual is considered a challenge and may lead to harmful consequences, whereas looking at the non-dominant - and non-threatening - individual is clearly a safer avenue⁵⁶. Assuming monkeys generalize this morphological characteristic of their own species (i.e. assuming that they exhibit a "simiomorphic" bias), part of their preference for trustworthy human faces may result from high FWHR signaling caution and low FWHR approachability. In other words, FWHR might be an objective feature kept through evolution to implicitly detect trustworthiness from faces.

Although this might seem to be a parsimonious interpretation, avoidance of faces with low FWHR is not sufficient to explain the differential attraction for trustworthy and non-trustworthy faces. Our correlation results show that FWHR accounts for 12.2% and 21.2% of the variance in looking in monkeys and humans, respectively. Furthermore, although monkeys tend to look longer at non-dominant than dominant human faces, this difference failed to reach significance. Thus other sort of cues must contribute to the monkeys' preference for trustworthy faces. Some of these are likely to be shared with facial cues for femininity, happiness or attractiveness, as evaluations of these social traits correlate strongly with trustworthiness judgments made by human subjects (Supplementary Table 2). For instance, macaque monkeys discriminate male versus female human faces⁵⁷. Thus we could speculate that cues to femininity also signal approachability, and combine with FWHR and other, yet to be determined physical features to generate the global impression of trustworthiness.

Our findings support the hypothesis that monkeys who are able to infer the "personality" of their conspecifics might recycle this measurable species-typical facial trait, for making similar inferences about human faces. In other words, FWHR might be an objective feature kept through evolution to implicitly detect trustworthiness from faces.

In humans, in addition to the objective FWHR feature a mechanism of emotion overgeneralization has also been considered important for trustworthiness judgments. According to the emotion overgeneralization hypothesis, resemblance of neutral faces to emotional expressions is perceived as indicating the trait attributes associated with these emotions^{13,16,58}. An emerging explanation for monkeys' preference for trustworthy human faces is that expertise with human faces enables them to detect gender and, possibly, the face general emotional valence, thus facilitating the perception of trustworthy faces as more positive and approachable than untrustworthy ones.

Using a set of standardized human faces, we have shown that macaques respond to facial human features linked to trust. Does this mean that monkeys are responding to the social trait of trust as humans we assume do? Monkeys' preference for looking at trustworthy more than

untrustworthy faces, and not the other way around, support this interpretation. Although the meaning of trustworthiness may be different between the two species as the presence of language in humans may further shapes the impression of trust and semantically enriches the concept in a categorical manner, here the behavioral response did not differ between the two species. We might assume that at an implicit level for both species a trustworthy face enhances approach behavior.

Because we used models of parametrized humans faces, it is unknown whether similar preferences would have been recorded if macaques were shown virtual parametrized monkeys faces. Given that cross interspecies abilities in the social domain are more difficult to prove, in the light of the present result, it is reasonable to assume that if macaques show spontaneous preference for human faces conveying trustworthiness they might also be able to do so for faces of conspecifics.

To our knowledge, data on what might constitute, for a monkey, a trustworthy or an untrustworthy conspecific's face, or whether facial features that convey this social trait in humans are present in monkey faces' and ecologically meaningful to a monkey observer, is not available yet. Investigating further whether monkeys exhibit first impression effects for trust in conspecifics' faces would be interesting, but quite challenging. Monkey faces generated with human-defined transformation rules may have very poor ecological validity. Yet, an interesting future study would be to manipulate fWHR of monkey face pictures and assess looking preference of monkey observers. However, as fWHR facial metric correlates with different social judgments⁵⁹, this approach may not fully capture the trustworthy features of monkeys' faces. A remaining option is to conduct an extensive ethological study aimed at identifying facial characteristics of more or less trustworthy/approachable individuals within a macaque social group, apply empirically derived transformation rules to generate an appropriate standardized set of macaque face stimuli and assess monkeys' viewing preferences for such stimuli.

Our results are also in line with a number of studies showing that macaques' social abilities go beyond their own species by encompassing the ability to understand interactive behaviors within the human repertoire. Monkeys can observe and interpret human social cooperation, by preferring to interact with individuals who demonstrate reciprocity with peers⁴². They also spend more time looking at humans that imitate their gestures (lip-smacking) but not at those that previously just stare in front of them⁶⁰. If monkeys can distinguish humans who reciprocate from those who don't, this suggests that they are attentive to visual social cues emitted by our species^{32,37,61}. Such comprehension of human social behavior might also be the basis of monkeys' ability to form human-like "first impression" of human faces as our results seems to indicate. Darwin proposed that facial displays of emotions serve to predict an individual's current intentions⁶². Inference of social trait is a cognitive mechanism that allows prediction of others' future behavior¹⁶. Invariant and morphological aspects of the face have a fundamental role in making these inferences. Considering the present findings, it is reasonable to assume that the implicit visual preference that monkeys and humans displayed is made possible thanks to a strong predisposition to use not only overt emotional cues but also stable face characteristics announcing covert social attitudes.

Physiognomy is the ancient art of connecting facial features with underlying character. It is unlikely and unexpected that judgments on social traits based on facial features are always accurate; however, there might be a reason why evolution is keeping the mechanisms necessary to be sensitive to trustworthiness facial features. Detecting fast who can be approached and who should be avoided may constitute a basic reflex-like mechanism intrinsically tied with all primates' social survival.

In non-human primates societies, group-living behaviors ensure protection from predators and improve access to food resources⁶³. Social individuals that reach a high level of exchange and support deal better with daily challenges maintaining a low physiological stress level⁶⁴. In anthropomorphic terms, sociable, confident and equable personality traits of non-human primates

are important characteristics to detect for boosting affiliative behaviors both with familiar and unfamiliar individuals.

Non-human primates are highly sensitive to, and use facial cues during social interaction. Evolutionary selection of trust mechanism might have its root on group formation. Natal dispersal, when animals migrate from their social group before they breed - a potentially relevant group stability factor⁶⁵⁻⁶⁷ - may provide a powerful evolutionary context for understanding why non-human primates are able to detect features of trustworthiness from faces. In a new group, animals are faced with unfamiliar individuals, in such case facial cues, facial expression and body size may assume greater importance for the survival, revealing with whom interact and whom should be avoided outside one's natal group. Thus, natal dispersal might justify the presence of face first impression mechanisms in monkeys, which might have been developed to cope with social unknown and unfamiliarity just like humans.

Materials and methods

Ethics statement

All experimental procedures were in accordance with the local authorities (Direction Départementale des Services Vétérinaires, Lyon, France) and the European Community standards for the care and use of laboratory animals [European Community Council Directive (1986), Ministère de l'Agriculture et de la forêt, Commission Nationale de l'Expérimentation Animale]. Human experiments were approved by the local ethical committee Sud-Est Centre Léon Berard , Lyon and sponsored by CNRS.

Experimental Design

Subjects. *Monkeys:* eight adult monkeys (Macaca Mulatta, one female and four males 4-17 years old, and Macaca fascicularis, three males 6 years old) have been tested. All animals were born in

outdoor enclosures and were then socially housed indoor, so they have been exposed to both conspecific and humans. *Humans participants:* fifty-four healthy subjects, (27 women, M age= 26.9 years, SD= 5.9), with normal vision, took part in the experiment. All were blind about the aim of the study; participants only knew to take part to a first impression study.

Stimuli. The stimuli used in the experiment were 48 computer-generated male faces created with the FaceGen software development kit (Singular Inversions, Toronto, Canada) selected from the Todorov's well-controlled quantitatively validated stimulus repertoire of faces. All faces were bald and Caucasian. The Trustworthiness database is composed of facial identities varying on 7 levels of trustworthiness⁴⁹. Todorov et al' work showed that human explicit judgments of trustworthiness match with the model's prediction⁴⁹ For the current study we selected from the 24 identities the two most extreme versions (-3 SD and +3 SD).

Task procedure. *Monkeys:* to assess preference formation, we used a preferential looking paradigm. For monkeys, pairs of faces were presented in a random order. Each face pair was presented twice to counterbalance for side of presentation. During the experiment, monkeys were seated in a primate chair inside a darkened room with their head restrained. Stimuli were presented on a 15-inch color monitor (1024 x 768 pixels) at a viewing distance of 24cm. A trial began with the appearance of a single fixation point in the center of the screen. Once the monkey fixated this point, two face stimuli subtending 13° x 21.2° of visual angle (207 x 340 pixels) were displayed and remained on the screen for up to 2s. The monkey was free to move its eyes over the images and received a juice reward provided its gaze stayed within the boundaries of the video monitor for the entire 2s period, otherwise the stimuli were extinguished and the trial discarded. Monkeys could choose to look outside of the face and still receive reward. The monkey's gaze position was monitored by ISCAN infrared eye tracking system at 200-Hz. Experimental control, stimulus presentation, data sampling and storage was done with REX/VEX software system. Before the

experiment, monkeys underwent a 5-point eye position calibration and were trained until they understood the visual exploration task using 8 different pairs of non-face biological and non-biological stimuli. *Humans participants*: healthy participants ($n=54$) were instructed to look at the same pairs of faces. We adopted the 5 seconds exploration duration in the human subjects experiment based on a similar study by Méary and colleagues (2014). This duration was reduced to 2 seconds for monkeys because preliminary testing showed that their attention decreased and gaze escaped the screen with longer durations thus making prolonged stimulus presentation worthless. Stimuli were displayed on a 17-inch computer screen at a resolution of 1280 x 1024 pixels using Presentation® software (Version 14.9, www.neurobs.com). The viewing distance from the participant's eyes to the screen on which stimuli subtending $7.8^\circ \times 12.5^\circ$ of visual angle (377 x 604 pixels) were displayed was 73cm. Humans' eye positions were recorded using an infrared video-based tracker (Tobii 1750) at a 60-Hz sampling rate and Clearview 2.7.0 allowed online recording of eye-gaze data. The two systems were synchronized using the Tobii extension for Presentation. In a second session the same pairs of faces were presented and humans were asked to explicitly select the most trustworthy faces. Eye positions were recorded until the response. During all sessions the experimenter monitored on-line the position of the subject's eye gaze that was projected on a second screen in the same room but placed far from the location of the participants. Prior to the experiment, humans underwent a 5 point-calibration task. The final experimental set comprised 48 trials.

Statistical Analysis

Pre-processing and data analysis.

ClearView fixation filter was used to filter the data for humans (with a visual angle of 1° and duration of 100ms). An in-house Matlab script was used to pre-process and filter monkeys' eye-tracking data. First, eye velocity for each location was computed as the angular distance traversed

by the eye within a 5ms moving window. Next, for each trial, a velocity threshold was set at three times the median during the 2s window. Data points that exceeded this threshold were considered as saccades. Fixation times were considered as the interval between two saccades with a minimum duration of at least 100ms, and fixation locations were defined as the eye position at the central fixation time point. In order to quantify allocation of attention to faces, regions of interest (ROI) delimiting each face were defined manually. The mean looking time was calculated as the average of the total time spent within each ROI during a trial. Only trials with at least one fixation at one of the two faces were included in the dataset. For the main statistical analysis, mean looking times on each face were calculated for each participant and for each trial.

Chance level analysis.

The chance level was computed as the ratio between the number of pixels on the ROI and total number of pixels on the screen. This ratio was then multiplied by the trial duration (2000ms for monkeys and 5000ms for humans), to obtain the fixation time that corresponds to random exploration of the whole screen. The chance level for monkeys was 160 ms and for humans was 853 ms. A t-test comparison of looking times was then performed to compare the chance level values for monkey and humans, to assess whether the real fixation time in the ROI vs the chance duration was significantly different.

Temporal dynamics and statistical analysis. In order to identify the time windows showing significant differences between trustworthy and untrustworthy faces, a large scale multiple testing procedures was designed. First, statistical differences between the number of fixations in trustworthy and untrustworthy ROIs were tested using T-tests at each time point. Then, multiple comparisons were performed using cluster-based permutation test (alpha cluster=0.05; number of permutation=100) ⁶⁸.

Spatial distribution of fixations. To provide information on the spatial distribution of the fixations, the barycenter of fixations and a heat map representation were calculated for each face at the subject level. Heat maps were calculated using Gaussian kernel density mapping of the fixations, weighted by the fixations' duration⁶⁹. Then, at the group-level, individual heat-maps were normalized and averaged to visualize the spatial distribution of the fixations of the studied population.

Facial width-to-height ratio (FWHR). To obtain a score for the facial width-to-height ratio, two independent raters measured the distance between the lip and brow (upper facial height) and the left and right zygion (bizygomatic width) of each face from the trust database. FWHR was calculated as width divided by height⁵³. Inter-rater reliability was high for all measures (all $R_s > .79$, all $P_s < 0.001$).

Acknowledgments

This research was funded by CNRS, ANR and by LABEX CORTEX (ANR-11-LABX-0042) University of Lyon, within the program "Investissements d'Avenir" (ANR-11-IDEX-0007) to AS and JRD. MC is supported by a French Minister of Research fellowship. We thank Andres Posada for help during eye movement analyses in human subjects and Dr. Suliann Ben Hamed for allowing testing one monkey.

Authorship and contribution.

AS proposed the study concept. All the authors designed the methods. Data collection was performed by MC and AG (human participants) and by EB (monkeys). Data analysis was conducted by MC, EB, AG and GL under the supervision of AS and JRD. MC, AG, AS and JRD wrote the paper. All authors discussed the findings and approved the final version of the manuscript for submission.

References

1. Cosmides, L. & Tooby, J. in *The adapted mind: Evolutionary psychology and the generation of culture* (ed. Barkow, Jerome H., Leda Ed Cosmides, and J. E. T.) 163–228 (1992).
2. Ross, W. & Lacroix, J. Multiple meanings of trust in negotiation theory and research: a literature review and integrative model. *Int. J. Confl. Manag.* **7**, 314–360 (1996).
3. Olivola, C. Y. & Todorov, A. Elected in 100 milliseconds : Appearance-Based Trait Inferences and Voting. *J. Nonverbal Behav.* 83–110 (2010). doi:10.1007/s10919-009-0082-1
4. Olivola, C. Y., Funk, F. & Todorov, A. Social attributions from faces bias human choices. *Trends Cogn. Sci.* 1–5 (2014). doi:10.1016/j.tics.2014.09.007
5. Willis, J. & Todorov, A. First Impression Making Up Your Mind After a 100-Ms Exposure to a Face. *Psychol. Sci.* **17**, 592–598 (2006).
6. Todorov, A., Olivola, C. Y., Dotsch, R. & Mende-Siedlecki, P. Social Attributions from Faces: Determinants, Consequences, Accuracy, and Functional Significance. *Annu. Rev. Psychol.* 1–46 (2015). doi:10.1146/annurev-psych-113011-143831
7. Dotsch, R. & Todorov, a. Reverse Correlating Social Face Perception. *Soc. Psychol. Personal. Sci.* **3**, 562–571 (2012).
8. Todorov, A., Pakrashi, M. & Oosterhof, N. N. Evaluating Faces on Trustworthiness After Minimal Time Exposure. *Soc. Cogn.* **27**, 813–833 (2009).
9. Weston, E. M., Friday, A. E. & Liò, P. Biometric evidence that sexual selection has shaped the hominin face. *PLoS One* **2**, (2007).
10. Stirrat, M. & Perrett, D. I. Valid facial cues to cooperation and trust: male facial width and trustworthiness. *Psychol. Sci.* **21**, 349–54 (2010).
11. Lefevre, C. E. *et al.* Facial width-to-height ratio relates to alpha status and assertive personality in capuchin monkeys. *PLoS One* **9**, e93369 (2014).
12. Oosterhof, N. & Todorov, A. The functional basis of face evaluation. *Proc. Natl. Acad. Sci. U. S. A.* **105**, 11087–11092 (2008).
13. Montepare, J. M. & Zebrowitz, L. A. Person perception comes of age: The salience and significance of age in social judgments. *Adv. Exp. Soc. Psychol.* **30**, 93–161 (1998).
14. DeBruine, L. M. Trustworthy but not lust-worthy: context-specific effects of facial resemblance. *Proc. Biol. Sci.* **272**, 919–22 (2005).
15. Oosterhof, N. & Todorov, A. Shared perceptual basis of emotional expressions and trustworthiness impressions from faces. *Emotion* **9**, 128–133 (2009).
16. Knutson, B. Facial expressions of emotion influence interpersonal trait inferences. *J. Nonverbal Behav.* **20**, 165–182 (1996).
17. Montepare, J. M. & Dobish, H. The contribution of emotion perceptions and their overgeneralizations to trait impressions. *J. Nonverbal Behav.* **27**, 237–254 (2003).
18. Todorov, A. Evaluating faces on trustworthiness: An extension of systems for recognition of emotions signaling approach/avoidance behaviors. *Ann. N. Y. Acad. Sci.* **1124**, 208–224 (2008).
19. Todorov, A., Baron, S. G. & Oosterhof, N. N. Evaluating face trustworthiness: A model based approach. *Soc. Cogn. Affect. Neurosci.* **3**, 119–127 (2008).
20. Van Essen, D. C. Organization of visual areas in macaque and human cerebral cortex. *Vis. Neurosci.* **1**, 507–521 (2004).

21. Tsao, D. Y., Moeller, S. & Freiwald, W. a. Comparing face patch systems in macaques and humans. *Proc. Natl. Acad. Sci. U. S. A.* **105**, 19514–19519 (2008).
22. Sereno, M. I. & Tootell, R. B. H. From monkeys to humans: What do we now know about brain homologies? *Curr. Opin. Neurobiol.* **15**, 135–144 (2005).
23. Sugita, Y. Face perception in monkeys reared with no exposure to faces. *Proc. Natl. Acad. Sci. U. S. A.* (2008). doi:10.1073/pnas.0706079105
24. Parr, L. a. *et al.* Experience-dependent changes in the development of face preferences in infant rhesus monkeys. *Dev. Psychobiol.* 1–17 (2016). doi:10.1002/dev.21434
25. Ferrari, P., Paukner, A., Ionica, C. & Suomi, S. J. Reciprocal Face-to-Face Communication between Rhesus Macaque Mothers and Their Newborn Infants. *Curr. Biol.* **19**, 1768–1772 (2009).
26. Deaner, R. O., Khera, A. V. & Platt, M. L. Monkeys pay per view: Adaptive valuation of social images by rhesus macaques. *Curr. Biol.* **15**, 543–548 (2005).
27. Johnson, M. H., Dziurawiec, S., Ellis, H. & Morton, J. Newborns' preferential tracking of face-like stimuli and its subsequent decline. *Cognition* **40**, 1–19 (1991).
28. Johnson, M. H. Subcortical Face Processing. **6**, 766–774 (2005).
29. Pascalis, O., de Haan, M. & Nelson, C. a. Is face processing species-specific during the first year of life? *Science* **296**, 1321–3 (2002).
30. Pascalis, O. & Kelly, D. J. The Origins of Face Processing in Humans. **4**, 200–209 (2009).
31. Quinn, P. C. *et al.* Looking across domains to understand infant representation of emotion. *Emot. Rev.* **3**, 197–206 (2011).
32. Gothard, K. M., Brooks, K. N. & Peterson, M. a. Multiple perceptual strategies used by macaque monkeys for face recognition. *Anim. Cogn.* **12**, 155–67 (2009).
33. Parr, L. a. The evolution of face processing in primates. *Philos. Trans. R. Soc. Lond. B. Biol. Sci.* **366**, 1764–1777 (2011).
34. Méary, D., Li, Z., Li, W., Guo, K. & Pascalis, O. Seeing two faces together: preference formation in humans and rhesus macaques. *Anim. Cogn.* **17**, 1107–19 (2014).
35. Sigala, R., Logothetis, N. K. & Rainer, G. Own-species bias in the representations of monkey and human face categories in the primate temporal lobe. *J. Neurophysiol.* **105**, 2740–52 (2011).
36. Dufour, V., Pascalis, O. & Petit, O. Face processing limitation to own species in primates: A comparative study in brown capuchins, Tonkean macaques and humans. *Behav. Processes* **73**, 107–113 (2006).
37. Leonard, T. K., Blumenthal, G., Gothard, K. M. & Hoffman, K. L. How macaques view familiarity and gaze in conspecific faces. *Behav. Neurosci.* **126**, 781–91 (2012).
38. Mosher, C. P., Zimmerman, P. E. & Gothard, K. M. Facial Expressions in Monkeys. **125**, 639–652 (2014).
39. Sliwa, J., Duhamel, J.-R., Pascalis, O. & Wirth, S. Spontaneous voice-face identity matching by rhesus monkeys for familiar conspecifics and humans. *Proc. Natl. Acad. Sci. U. S. A.* **108**, 1735–1740 (2011).
40. Meltzoff, A. N. & Moore, K. M. Imitation_of_facial_and_manual_gestures.pdf. *Science (80-)*. **198**, 75–78 (1977).
41. Ferrari, P. *et al.* Neonatal imitation in rhesus macaques. *PLoS Biol.* **4**, 1501–1508 (2006).
42. Anderson, J. R., Kuroshima, H., Takimoto, A. & Fujita, K. Third-party social evaluation of

- humans by monkeys. *Nat. Commun.* **4**, 1561 (2013).
43. Anderson, J. R., Takimoto, A., Kuroshima, H. & Fujita, K. Capuchin monkeys judge third-party reciprocity. *Cognition* **127**, 140–6 (2013).
 44. Paukner, A., Suomi, S. J., Visalberghi, E. & Ferrari, P. F. Capuchin monkeys display affiliation toward humans who imitate them. *Science* **325**, 880–883 (2009).
 45. Braccini, S. N., Lambeth, S. P., Schapiro, S. J. & Fitch, W. T. Eye preferences in captive chimpanzees. *Anim. Cogn.* **15**, 971–978 (2012).
 46. Tsao, D. Y., Freiwald, W. A., Knutsen, T. A., Mandeville, J. B. & Tootell, R. B. H. Faces and objects in macaque cerebral cortex. *Nat Neurosci* **6**, 989–995 (2003).
 47. McMahon, D. B. T., Jones, A. P., Bondar, I. V. & Leopold, D. A. Face-selective neurons maintain consistent visual responses across months. *Proc. Natl. Acad. Sci.* **111**, 8251–8256 (2014).
 48. Sliwa, J., Plante, A., Duhamel, J. R. & Wirth, S. Independent Neuronal Representation of Facial and Vocal Identity in the Monkey Hippocampus and Inferotemporal Cortex. *Cereb. Cortex* **26**, 950–966 (2016).
 49. Todorov, A., Dotsch, R., Porter, J. M., Oosterhof, N. N. & Falvello, V. B. Validation of data-driven computational models of social perception of faces. *Emotion* **13**, 724–38 (2013).
 50. Hehman, E., Leitner, J. B., Deegan, M. P. & Gaertner, S. L. Picking Teams: When Dominant Facial Structure is Preferred. *J. Exp. Soc. Psychol.* **59**, 51–59 (2015).
 51. Mileva, V. R., Cowan, M. L., Cobey, K. D., Knowles, K. K. & Little, a. C. In the face of dominance: Self-perceived and other-perceived dominance are positively associated with facial-width-to-height ratio in men. *Pers. Individ. Dif.* **69**, 115–118 (2014).
 52. Costa, M., Lio, G., Gomez, A. & Sirigu, A. How components of facial width to height ratio differently contribute to the perception of social traits. *PLoS One* **12**, 1–12 (2017).
 53. Carré, J. M. & McCormick, C. M. In your face: facial metrics predict aggressive behaviour in the laboratory and in varsity and professional hockey players. *Proc. Biol. Sci.* **275**, 2651–6 (2008).
 54. Borgi, M. & Majolo, B. Facial width-to-height ratio relates to dominance style in the genus *Macaca*. *PeerJ* **4**, e1775 (2016).
 55. Shimojo, S., Simion, C., Shimojo, E. & Scheier, C. Gaze bias both reflects and influences preference. *Nat. Neurosci.* **6**, 1317–1322 (2003).
 56. Maestripietri, D. & Wallen, K. Affiliative and submissive communication in rhesus macaques. *Primates* **38**, 127–138 (1997).
 57. Afraz, A., Boyden, E. S. & DiCarlo, J. J. Optogenetic and pharmacological suppression of spatial clusters of face neurons reveal their causal role in face gender discrimination. *Proc. Natl. Acad. Sci.* **112**, 6730–6735 (2015).
 58. Todorov, A. Evaluating faces on trustworthiness. *Ann. N. Y. Acad. Sci.* (2008).
 59. Geniole, S. N., Denson, T. F., Dixon, B. J., Carré, J. M. & McCormick, C. M. Evidence from meta-analyses of the facial width-to-height ratio as an evolved cue of threat. *PLoS One* **10**, 1–18 (2015).
 60. Paukner, A., Huntsberry, M. E. & Suomi, S. J. Visual discrimination of male and female faces by infant rhesus macaques. *Dev. Psychobiol.* **52**, 54–61 (2009).
 61. Dahl, C. D., Wallraven, C., Bülthoff, H. H. & Logothetis, N. K. Humans and macaques employ similar face-processing strategies. *Curr. Biol.* **19**, 509–13 (2009).

62. Darwin, C. *Expression of the Emotions. Man and Animals* (1872).
63. McCowan, B. *et al.* Network stability is a balancing act of personality, power, and conflict dynamics in rhesus macaque societies. *PLoS One* **6**, e22350 (2011).
64. Brent, L. J. N., Semple, S., Dubuc, C., Heistermann, M. & MacLarnon, A. Social capital and physiological stress levels in free-ranging adult female rhesus macaques. *Physiol. Behav.* **102**, 76–83 (2011).
65. Beisner, B. A., Jackson, M. E., Cameron, A. & Mccowan, B. Effects of natal male alliances on aggression and power dynamics in rhesus macaques. *Am. J. Primatol.* **73**, 790–801 (2011).
66. Ekernas, L. S. & Cords, M. Social and environmental factors influencing natal dispersal in blue monkeys, *Cercopithecus mitis stuhlmanni*. *Anim. Behav.* **73**, 1009–1020 (2007).
67. Weiß, B. M. *et al.* Individual dispersal decisions affect fitness via maternal rank effects in male rhesus macaques. *Nature* **6**, 32212 (2016).
68. Maris, E. & Oostenveld, R. Nonparametric statistical testing of EEG- and MEG-data. *J. Neurosci. Methods* **164**, 177–190 (2007).
69. Caldara, R. & Miellet, S. i Map : a novel method for statistical fixation mapping of eye movement data. *Behav. Res. methods* (2011). doi:10.3758/s13428-011-0092-x
70. Oosterhof, N. N. & Todorov, A. The functional basis of face evaluation. *Proc. Natl. Acad. Sci. U. S. A.* **105**, 11087–92 (2008).

Supplementary Material

Supplementary Table 1. Monkeys' individual biographical and behavioral characteristics.

	Age at the time of the experiment	Sex	Specie	Total Looking time at the faces	Total time looking at Trustworthy face	% bias for trustworthy face	p value
Y	15 years	M	Rhesus Macaque	657,29	508,6	+27,4	0.004
T	6 years	M	Macaque fascicularis	870,13	635,3	+23.0	0.001
O	17 years	M	Rhesus Macaque	1282,6	902,5	+20.4	0.006
S	6 years	M	Macaque fascicularis	308,15	203,2	+15.9	0.18
E	6 years	M	Macaque fascicularis	726,18	476,3	+15.6	0.05
V	5 years	M	Rhesus Macaque	885,2	503,3	+6.9	0.24
D	4 years	M	Rhesus Macaque	468,43	243,1	+1.9	0.83
Z	13 years	F	Rhesus Macaque	1245,95	630,8	+0.6	0.93

Supplementary Fig. 1. Human subjects: correlation between difference in looking time (trustworthy - untrustworthy) during the implicit and the explicit task. We observed a positive correlation ($r=0.30$; $P=0.027$). Subjects that looked more the trustworthy faces during the implicit task also looked more the same faces when they had to explicitly select the trustworthy face. This result supports that eye movements in humans predict their explicit judgments.

Dominance

To demonstrate monkeys' selective preference for trustworthy faces we designed a new experiment where we tested the animals' perception of dominant/submissive faces. As it was the case in the trust experiment we presented dominant and subdominant faces taken from Todorov's database. The stimuli used were 50 computer-generated male faces created using the software FaceGen Modeller (<http://facegen.com>, version 3.1). Each facial identity varied along the dimension of dominance on both shape and reflectance. The Dominance database is composed of facial identities manipulated to create seven versions (-3 SD, -2 SD, -1 SD, 0 SD, 1 SD, 2 SD, and 3 SD) on dominance (Todorov et al. 2013). For the current study, we selected from the 25 identities the two most extreme versions (-3 SD and +3 SD), resulting in 25 pairs. The same monkeys ($N=8$) performed the preferential looking paradigm. Monkeys spent 574,51ms (SD=193,88) looking at submissive human faces and 447,01 ms (SD=209,33) looking at the dominant ones, but the difference between the two types of face was not significant (paired sample t-test; $t_{(7)}=1.66$; $P = 0.13$). This result suggests that monkeys' do not distinguish between dominant and subdominant human faces. We can add these new results in the supplementary material.

Correlation between human looking time, happiness and femininity

Perceived happiness and femininity of face stimuli contribute to trustworthiness judgments⁷⁰. This was checked through a quantification of these two attributes in two separate experiments, by independent human observers ($N=7$) in the following manner. Pairs of randomly selected faces of all identities were presented on the screen and subjects had to choose either the most 'happy' or the most 'feminine' one by pressing a key on a keyboard. Each face stimulus was presented at least 16 times by subject (1225 trials). Happiness and femininity scores for each face were given by the percentage of instances the stimulus was selected as the happiest and the most feminine, respectively. Mean inter-subject reliability in these evaluations was high for happiness, $r_{(48)} = 0.88$ and for femininity, $r_{(48)} = 0.60$. We computed correlation between happiness, femininity and FWHR

and explicit trustworthiness judgments (value used: -3; +3). As expected, all correlations were significant: happiness, $r_{(48)} = 0.91$ ($P < 0.000001$); femininity, $r_{(48)} = 0.88$ ($P < 0.000001$); FWHR, $r_{(48)} = -0.54$ ($P < 0.00005$). Correlation between happiness and femininity was significant $r_{(48)} = 0.76$ ($P < 0.000001$); correlation between happiness and FWHR was significant $r_{(48)} = -0.46$ ($P < 0.0005$). At the same way femininity was correlated with FWHR $r_{(48)} = -0.60$ ($P < 0.00001$) (Supplementary Table2).

To establish whether humans' looking preference was affected by happiness and femininity we performed further correlation analyses using the he obtained scores of these attributes and subjects' looking time. We found that humans' total viewing time on a face was positively correlated to the emotion score (happiness: $r_{(48)} = 0.731$, $P < 0.000001$) and to the face femininity score: ($r_{(48)} = 0.738$, $P < 0.000001$). These results confirm that human observers perceive trustworthy faces as happier and more feminine than untrustworthy faces.

Supplementary Table 2. Pearson coefficient correlation analyses among all variables. Trustworthiness values used for these correlations are -3/+3. Happiness and femininity scores for each face were given by the percentage of instances the stimulus was selected as the happiest and the most feminine. FWHR score have been obtained from the measures of two independent raters.

r	Trustworthiness	Happiness	Femininity	FWHR
Trustworthiness	1			
Happiness	0,91	1		
Femininity	0,88	0,76	1	
FWHR	-0,54	-0,46	-0,60	1

RESPONSES TO REVIEWERS

Reviewer #1 (Remarks to the Author):

Costa and colleagues investigate whether monkeys show a preference for viewing faces that is dependent on their perceived trustworthiness (based on prior studies that have investigated this characteristic in humans). They find that both monkeys and humans exhibit a preference to look at more trustworthy faces. Further, monkeys showed an upward eye movement to the eye region when viewing trustworthy faces, which the authors interpret as an approach behavior. Finally, they report correlations between the width-to-height ratio of the faces and the looking time in both monkeys and humans. This is a very interesting comparative study that suggests a common basis for evaluation of faces in monkeys and humans. In general, the experiment is well-conducted, with data from a reasonably large number of monkeys ($n = 8$) and the manuscript well-written. But I think there are some specific concerns the authors need to address to strengthen their findings.

- 1) It's a little concerning that the stimulus duration was not matched between species (humans, 5 s; monkeys 2 s) and it can't be ruled out that some of the differences between the species results from this change. I would strongly recommend the authors collect some additional human data with the same presentation duration as the monkeys and see if duration effects any of the patterns observed.*

Response: We thank the Reviewer for the interest she/he expresses on our findings. To address this specific point, we ran a new experiment using a 2 seconds stimulus exposure duration. Using a G-power analysis technique, we computed an *a priori* sample size N as function of the effect size in the 2 seconds experiment with monkeys ($d_z=1.164$) and in the 5 second experiment with humans ($d_z=0.615$ with power=0.95 and significance level =0.05, one-tailed). The total sample size was 10 and 30 respectively. We therefore tested 20 human participants using 2 seconds stimulus duration design and replicated the main result showing a significant preference for trustworthy faces. To be consistent with the procedure used in the monkey experiment, as suggested by the Referee, we now report these new data in the main text.

The human data reported in the original version of the manuscript obtained on 54 subjects with a 5 seconds stimulus duration have been moved to the supplementary material as a preliminary experiment. This was done in order to maintain the comparison between visual preference in the implicit task and in the explicit trustworthiness judgements already obtained from the same subjects (See p.35).

The new results are described in the main text at page 7

Humans followed the same pattern, spending most of the time looking more to faces ($AIC= 4.43 \cdot 10^5$) than predicted by a central bias model ($AIC=4.86 \cdot 10^5$) (see supplementary method and supplementary Fig.1). Importantly, humans showed a significant bias in favor of the trustworthy (mean \pm s.d = 865.35 ± 120.44 ms) stimuli compared to the untrustworthy ones (mean \pm s.d = 796.35 ± 105.00 ms) (paired sample T-test; $t(19) = -1.87$, $P < 0.05$, $\eta^2 = 0.15$) (Fig.1d).

In accordance with the new findings, Figure 1 has been changed as following:

Fig. 1. Looking preference for trustworthy vs. untrustworthy faces by rhesus macaques and human subjects. MONKEYS (N=8): (A) Mean looking time in milliseconds (ms) for the most trustworthy (+3SD of the neutral face) and the least trustworthy (-3SD of the neutral face) versions of the same facial identities. Circles indicate individual data points. The error bars denote standard error of the mean. * $P < 0.05$. Monkeys looked significantly longer at the two faces than predicted by chance and looked more at trustworthy than untrustworthy faces. (B) Time course of looking preference. Mean viewing time ratio between each facial prototype. A cluster-based permutation test showed that preference for the trustworthy faces (green line) was significant between 510ms and 1485ms ($P < 0.05$ corrected for multiple comparison). (C) Gaze heat maps for trustworthy and untrustworthy faces averaged across subjects (trustworthy face on the left by convention, facial prototype spatial location was counterbalanced within and between subjects). Yellow dots show fixation centers of gravity for each subject. **HUMANS (N=20)** (D-E-F) Plots show (D) significantly longer mean looking times at trustworthy than untrustworthy faces and (E) onset of preference for trustworthy faces (200ms to 1152ms). Note that the average barycenter of fixation was located in the region surrounding the nose in monkeys whereas it is around the eye and nose region in humans (C, F).

2) Chance level was computed as the ratio between the number of pixels on the ROI and the total number of pixels on the screen. I think this may be a misleading estimate since there's a well-known central bias for fixations (i.e. people do not make fixations near the edges of the screen/stimulus). I'm not sure of a good solution to this concern - could the authors use eye

movement data from scene stimuli to estimate what the effective potential fixation area of the screen is and use this value as the denominator in their ratio?

Response: We agree with the Reviewer and following her/his remark we propose a new way to analyze chance level that in our opinion takes into account the central bias problem. We investigated visual exploration density over the screen using three models (central bias Gaussian model, two faces Gaussian model, two faces and central bias Gaussian model) based on the Akaike information criterion where the best model is the one with the lowest AIC value. We found that monkeys' and humans' visual exploration was better explained by the “two faces and central bias Gaussian model”.

The procedure is described in the Supplementary section at page 18:

Visual exploration density: across models comparison using the Akaike Information Criterion (AIC). To estimate whether monkeys and humans preferentially gazed within the faces' area compared to the rest of the screen (see Supplementary Fig. 1 for the analysis on density of fixations over the entire screen), we submitted participants' exploration density over the screen to three Gaussian-mixture models, based on a selection procedure applying the Akaike information criterion (AIC). The first model, a “central bias Gaussian model”, represents density of exploration as a Gaussian function centered on the screen, and therefore this is the model that best fits with the central bias behavior usually shown by humans while looking at scenes. The second, a “two faces Gaussian model”, density of exploration is exemplified by two Gaussian functions, each centered on one of the regions of interest (Left or Right face). Finally, the third, a “two faces and central bias Gaussian model”, estimates density exploration behavior by combining the two previous models. We considered as the best model the one that reports the lowest AIC values because of the quality of the fit and the complexity of the model. In monkeys, (Supplementary Figure 1), the best model was the “two faces and central bias Gaussian model” (AIC= $6.44 \cdot 10^5$) compared to the “central bias Gaussian model” (AIC= $6.47 \cdot 10^5$). In humans, a similar result was found (“two faces and central bias Gaussian model” with AIC= $4.43 \cdot 10^5$ while “central bias Gaussian model” reached a higher AIC= $4.86 \cdot 10^5$).

The following figure has been added in the Supplementary material:

Supplementary Fig. 1. Monkeys and Humans density of exploration over the screen and across models comparison scores using the Akaike Information Criterion (AIC). Left upper and lower panel: Mean density of exploration in monkeys (A) and humans (D) over the whole screen (yellow-orange indicates high density exploration, dark blue low density exploration, rectangles indicates position of faces on the screen). Middle upper and lower panel: Projection of mean density of exploration on the X-axis of the screen in monkeys (B) and humans (E) (with 95% confidence interval). Right upper and lower panel: AIC value according to the central bias Gaussian model (red), two faces Gaussian model (yellow), two faces and central bias Gaussian model (green) in monkeys (C) and humans (F).

In the main text at page 6-7, we have replaced the paragraph regarding chance level analysis with the following ones:

The first analysis, as expected, revealed that monkeys were attracted to both faces, spending more time on these stimuli (Akaike Information Criterion (AIC)= $6.44 \cdot 10^5$) than predicted by a central bias model (AIC= $6.47 \cdot 10^5$, see supplementary method and supplementary Fig.1).

Humans followed the same pattern, spending most of the time looking more to faces (AIC= $4.43 \cdot 10^5$) than predicted by a central bias model (AIC= $4.86 \cdot 10^5$) (see supplementary method and supplementary Fig.1).

3) *The introduction and discussion are very long and quite meandering. The manuscript would benefit from a much tighter introduction that references the essential background only, and a discussion that minimizes the currently extensive speculation and stays closer to the data.*

Response: Following the reviewer suggestion we removed several paragraphs from both the introduction and the discussion. We choose to not show these suppressions in the final revised main text to facilitate the reviewers reading. However, if the reviewer wishes to see them we would be happy to provide another version where these changes are visible.

4) *The authors cite work by Sugita suggesting that monkeys deprived from seeing faces still show a preference for faces over other objects. The authors might also want to consider the work from Marge Livingstone's lab (e.g. Arcaro et al, 2017) that suggests a slightly different picture.*

Response: We thank the reviewer for suggesting a more balanced view on the role of innate vs acquired experience on face perception. We now discuss the interesting work of *Livingstone and colleagues* at page 4.

Yet, as proposed by Livingstone and colleagues, extensive exposure to faces may still be necessary for neonatal macaques to discriminate between faces and for the proper functional specialization of the visual system to emerge²⁸

5) *I highly recommend the authors show the data points/lines for all 8 monkeys individually in some of their figures (e.g. Figure 1A, Figure 2).*

Response: We add data points for both monkeys and humans, see Figure 1A and 1D

6) Do humans show any specific bias similar to the monkeys' upward fixations for trustworthy faces on the second saccade?

Response: We performed this analysis with the new dataset of 20 human subjects but we did not find the same bias as reported for monkeys. This might be explained by the fact that humans' attention to face is by default directed towards the eye region. We add this information in the main text at page 9:

We performed the same analysis (with 3 fixations) in humans. There was no main effect of the face category ($F(1,19)=0.015$, $p=0.90$), and no significant interaction between face category and fixation order ($F(2,38)=0.369$, $p=0.69$) but a main effect of fixation order ($F(2,38)=4.664$, $\eta^2=0.197$, $p=0.015$). This result is however not surprising given that spontaneous humans' gaze is primarily directed towards the eyes region (overall Y coordinate was centered on the eyes for both face type (mean Y coordinate= 460.0 ± 5.6)).

7) In Figure 2 it would be helpful to provide some indication of how the y coordinate values correspond to a face stimulus. How high was the face? Is a value of 50 a large difference in location? Perhaps the authors could show a face to the left of the y-axis to provide some measure of scale relative to the stimulus.

Response: We now provide as an example the trajectories of the first two saccades from Monkey 1 showing the shift on the y-axis for trustworthy and untrustworthy faces. This figure now appears next to the original Figure 2 where we reported the graph showing the magnitude of the shift in the monkey group (Figure 2A-B).

Fig. 2. Fixations sequence analysis. **A.** Graph shows monkeys' Y coordinates of first and second fixations weighted by duration (first in blue, second in yellow) and type of face (trustworthy in green, untrustworthy in red); error bars show standard deviations. Fixations closer to the eye region are closer to $Y=300$. The location of the first fixation on the face is not different for the trustworthy and untrustworthy face but the location of second fixation is closer to the eye region only for the trustworthy face suggestive of an approach behavior. **B.** Position of the first (in blue) and second fixations (in yellow) over the trustworthy face (left, green ROI) and untrustworthy faces (right, red ROI) for monkey 1.

8) *In the second sentence of the Discussion the authors highlight the results of the mean duration of the first fixation analysis. But this analysis was only conducted for monkeys that showed a preference based on trustworthiness (or a trend) and I think the description is potentially misleading without the context. Moreover, I'm also not entirely convinced that limiting the analysis in this way is appropriate.*

Response: We agree with the reviewer and following her/his suggestion we have removed this analysis as it does not add any supplementary information.

9) *Page 13: "Since we presented two faces simultaneously without giving explicit instructions to participants, it is not surprising that human preference for trustworthy faces emerged later in time".*

Response: In the light of the new data in human subjects this conclusion is obsolete.

10) *Page 19, Stimuli: "For the current study we selected from the 24 identities the two most extreme versions (-3SD and +3SD)". This sentence is unclear. Presumably, the authors mean they selected the 24 most trustworthy and the 24 least trustworthy?*

Response: Thanks for pointing to the lack of clarity of this sentence which has been rewritten as following (see page 16):

For the current study, we selected the most (+3 SD, $N=24$) and the least (-3SD, $N=24$) trustworthy faces. On each trial a couple of the same identity differing only for their level of trustworthiness - associated features was presented.

Reviewer #2 (Remarks to the Author): *This paper reports a novel and quite surprising finding: macaque monkeys show a preference for trustworthy-looking human faces. It is well established that people hold specific stereotypes about trustworthy appearance and act on these stereotypes. The faces used by the authors were generated by a model that visualizes these stereotypes. The*

finding that macaques show a preference similar to humans is surprising, because the facial cues are fairly subtle (the fWHR is a very crude cue that happened to correlate with judgments of trustworthiness; see below).

The authors should take a look and cite a highly relevant paper by Jessen & Grossman (2016). Journal of Cognitive Neuroscience, 28, 1728-1736.

These authors used the same preferential looking paradigm and the same stimuli to study the behavior of 7-month old human infants. The findings are remarkably similar to the present findings: infants prefer to look at trustworthy than at untrustworthy looking faces, but show no such discrimination for dominant vs. submissive looking faces. These findings provide additional credence to the present findings. Yet it is easier to explain the infants than the macaque's findings. By 7-months of age, infants can discriminate positive and negative expressions of emotions and are most likely to have a woman as a primary caregiver: cues that are correlated with perceptions of trustworthiness. It is harder to make the same case for monkeys unless one invokes some sort of social learning from observing humans. The authors seem to invoke a nativist explanation but then it is important to show that the same configurations of features in monkey and human faces trigger similar perceptions of approach/avoidance.

Response: We thank the reviewer for the suggestion. This reference is now quoted at page 4, in the following sentences.

Remarkably, it has been shown that 7-month old human infants show a looking preference for trustworthy faces compared to the untrustworthy ones, while such sophisticated discrimination is not found for dominant vs submissive faces³⁵.

I would also suggest looking at the recent work of Margaret Livingston from Harvard Medical School. Her group raised monkeys without visual exposure to faces (similar to Sugita) and then used fMRI to search for face selective regions. The findings suggest that such regions are not formed unless monkeys are exposed to faces. This suggests that one needs an extensive learning and exposure to faces to develop the proper specialization and sensitivity to minor differences between different faces. This learning interpretation doesn't take anything away from the originality of the authors' findings and needs to be discussed as a plausible mechanism.

Response: We thank the reviewer for the suggestion. We now report the interesting work of Livingstone and colleagues at page 4. Please see also point 4 of Reviewer 1.

Looking at Suppl Table 1, the youngest monkey was 4 years old and the oldest 17. Unfortunately, the sample is too small to estimate the relationship between age as a proxy for experience with humans and bias to look at trustworthy faces (nevertheless, a quick analysis shows a correlation of .36; the scatter plot is reasonable with one outlier).

Response: Following the reviewer's suggestion we add a new supplementary figure (page 30) showing the bias on trustworthy faces fitted against monkeys' age. We also add this information in the main text at page 9.

Correlation between monkeys' age and the percentage of bias toward trustworthy-associated facial cues

It is reasonable to assume that preference toward trustworthy-associated facial cues is shaped by experience. Age was selected as an indicator of monkey's expertise in interacting with humans and correlated to the percentage of bias toward trustworthy-associated facial cues. The overall

correlation did not reach significance ($r(8)=0.365$ unilateral, $p=0.187$), but when we excluded the female outlier of the correlation, we found a positive correlation between age and preference for trustworthy-associated facial cues ($r(7)=0.675$, unilateral, $p=0.048$, without monkey Z), see Supplementary Fig.2.

Supplementary Fig. 2. Correlation between monkeys' age and percentage of bias toward trustworthy associated-facial cues. (Left) Percentage of bias for trustworthy-associated facial cues ($r(7)=0.67$, $P<0.05$) positively correlated with the age of the monkey, i.e., the older the monkey the larger the bias toward trustworthy-associated facial cues. Each point corresponds to a monkey. For this analysis the only female monkey outlier of the group was excluded though we left her position on the plot (in red) for illustration purposes.

The authors invoke the FWHR as a possible universal feature: "FWHR might be an objective feature kept through evolution to implicitly detect trustworthiness from faces." However, as they point out, this feature accounts for very little of the variance of trustworthiness judgments. Initially, this feature was interesting to psychologists and evolutionary biologists because of its possible sexual dimorphism, but recent meta-analyses show that its correlation with gender is minuscule relative to correlations with body/height and upper body strength. In humans, the measure is also correlated with body-mass index and this correlation can explain much of the observed effects in the literature.

The overgeneralization mechanism can serve as a potential explanation of the findings if monkeys are sensitive to emotional expressions in humans; presumably, more caring humans would have more positive expressions when handling the monkeys and this might be consistent across situations. This seems to be consistent with the fixation patterns of the monkeys (Fig. 1C), patterns clustered around the nose/mouth. In sum, sound research with an interesting and provocative finding in need of a good explanation.

Response: Thank you for these comments. We agree with the reviewer about the importance of the overgeneralization mechanisms to explain the preference we found here in monkey. Nevertheless, we believe that the observed effect is multifactorial. As argued in the discussion the stimuli used in

this experiment don't allow a firm conclusion on this point. In a future study, it would be interesting to create stimuli controlling for FWHR and emotional features.

Reviewer #3 (Remarks to the Author) (See also attached PDF) *General comments: The manuscript reports findings from a comprehensive comparative study investigating implicit preference for human faces characterised by anatomical features associated with trustworthiness. The findings show that macaques show a preference for trustworthy human faces, but with a slightly different pattern of attention to humans. The paper is well written, clear and complete. The design is appropriate, and the results seem strong. I have to admit that I am not very familiar with the analysis of eye tracking data, but they seem sound. I like the direct comparison between macaques and humans using the same methods.*

One addition that could be envisaged is some analyses of pupil dilation, which would the authors to have stronger interpretation of the attention pattern. If pupil dilation, and therefore arousal, is higher when looking at untrustworthy faces this could suggest that macaques avoided them. If pupil dilation was higher for trustworthy faces this would strengthen the approach interpretation.

Response: We thank the reviewer for this suggestion and ran this analysis. We found that human subjects constricted their pupil when looking at faces compared to the rest of the screen, evidently driven by the brightness difference between the stimuli and the display screen's darker background and a possible additional attentional component as previously observed by other studies (see references 68 and 69 in the main text), but no effect of trustworthiness associated cues has been found. Unfortunately, pupil size data was not recorded in the monkey subjects.

The analysis on pupil size is now reported in the supplementary section at page 33:

Pupil size in humans (n=20)

Pupil size data was not recorded in monkey subjects. In human, to assess whether a difference in pupil dilation existed between the two conditions (trustworthy –untrustworthy) we analyzed pupil data in our participants. First, extracted pupil data (pupil size in pixels) was linearly interpolated to fill missing values from blinks. Then, pupil data were band-pass filtered (between 0.175Hz and 3.5Hz) for each participant separately, on his/her entire time course to remove the slow derivation of pupil size and some transient artifacts that may appear in measurements^{1,2}.

During the task, participants' mean pupil size measured before filtering was 3.7mm (SD=0.67). Then, for each participant, the filtered pupil size values were extracted whenever the gaze was in the trustworthy-face region of interest and averaged across trials. The same procedure was used to calculate the magnitude of pupil size modification over untrustworthy faces. First, we tested whether pupil size was different from the average pupil size when gaze was over the face area and second, if pupil size was different over the two regions of interest (trustworthy vs untrustworthy). Results indicate that humans constricted their pupil both when they gazed at trustworthy and untrustworthy faces (trustworthy: mean $-0.041 \pm s.d=0.026$, $t(19) = -6.72$, $P < 0.001$; untrustworthy: mean $\pm s.d = -0.038 \pm 0.02$, $t(19) = -7.16$, $P < 0.001$), an effect most likely driven by attention^{1,2} although a brightness difference between the face stimuli and the dark background of the visual display cannot be excluded. No significant differences were found between the two faces category (paired sample T-test; $t(19) = 0.087$, $p=0.196$).

Overall, the interpretations of the results are suitable, but I would like to see a bit more caution at times and more consistency in the use of some key terms. The discussion is interesting but features

quite a few repetitions, which makes it quite lengthy. Perhaps this could be reorganised in a more concise way. I believe the contribution of this paper is worthy of publication in Nature Communications, provided that the authors make a few changes. Given the comparative nature of the study and the methods used, I think this work will be of interest to a broad readership.

Detailed comments There were no line numbers so see the attached file for detailed comments.

Response: We thank the reviewer for the detailed comments. We now add line numbers and we made the following changes based on her/his comments

This is all clear but in addition, it would be nice to have a sentence about the possible function of these first impressions.

We add a sentence at page 3 about the possible function of first impression mechanism:

A possible function of face first impression is to provide a sort of others' social identikit to facilitate decisions, like approaching or avoiding unfamiliar individuals, choosing a candidate during the voting process etc.^{9,10}

The results are only similar if dominance can be linked to trustworthiness. I don't think there is any evidence for that. I would rephrase this.

Response: Line 51. We changed “similar results” with “related results”

I don't think 'personality' is the correct word here. Dominance is not a personality trait but an attribute of repeated interactions between 2 individuals, characterised by a consistent outcome. You could use 'assertive' as in the original paper.

Response: Line 52-56. We rephrased the sentence as follow:

This raises the possibility that species-typical facial traits are reliable cues used by monkeys to infer conspecifics' self-confidence.

Do these results hold when controlling for FWHR? If so, it would be worth mentioning. (It has been shown that the facial width-to-height ratio (FWHR)¹¹, a morphometric measure that relies on face structure, predicts explicit judgments of trustworthiness¹².)

Response: We thank the reviewer to raise this interesting point. FWHR is a morphological cue that correlates with social traits. It may be interesting to build stimuli that vary on femininity or trustworthiness cues controlling for FWHR and test whether the perception of these social traits change comparing to a condition where FWHR is not controlled. To our knowledge, no studies have directly tested this issue.

*This doesn't really justify the use of the preferential looking paradigm. It justifies looking at attention to faces. I think the rationale for using this approach could be explained better here. A useful reference for this would be: Winters, S., Dubuc, C., & Higham, J. P. (2015). Perspectives: The looking time experimental paradigm in studies of animal visual perception and cognition. *Ethology*, 121(7), 625-640.*

Response: Line 71-72. We now provide a rationale for the use of the preferential looking paradigm by referring to the work of Winters et al, 2015. We added the following sentence.

This approach is relevant for studying sensitivity to trustworthiness-associated cues, gaze direction and visual exploration strategies in both species²³.

In addition to these studies, there is a recent paper showing attention to face-like stimuli in-utero: Reid, V. M., Dunn, K., Young, R. J., Amu, J., Donovan, T., & Reissland, N. (2017). The human fetus preferentially engages with face-like visual stimuli. Current Biology, 27(12), 1825-1828. This could be mentioned here.

Response: Line 81. We now report the study of Reid et al 2017 showing attention to face-like stimuli in utero.

Response: Line 83 We also report the study of Jessen & Grossman 2016 showing preference for trustworthy faces in 7-months old human infants and no discrimination for dominant faces.

Response: Line 78-80 We now included the results of Livingstone as suggested by reviewer 1 and reviewer 3

This phrasing is awkward. I would simply say 'support'

Response: Line 88. We changed “be advocated in favor of” with “can support”

Rather than differing in trustworthiness, they differ in anatomical features associated with differing levels of trustworthiness. I think this is quite important and the authors should be consistent in their use of the term. I prefer 'trustworthiness-associated features', as used in the abstract. The authors could also say earlier in the manuscript that this is what they refer to when they write 'trustworthiness'.

Response: Line 109. We now use trustworthiness associated-features as suggested by the reviewer.

From the figure it looks like a significant part of the fixations were also around the nose.

Response: Line 166. We added that human subjects' fixations were also around the nose region.

I think this needs some explanations. What is the support for this interpretation?

Response: Line 177-178. We have re-phrased the sentence:

This hypothesis is in line with previous studies showing that prolonged eye contact in great apes signals mild threat, while gaze avoidance indicates submission^{50,51}.

This is not necessary. Instead, I would add the sample size in the previous sentence.

Response: We removed this analysis, please see also point 8 of reviewer 1.

Here and throughout I think the authors should be more cautious about the link between attention pattern and approach/avoid behaviour, unless they can provide strong evidence supporting this interpretation. It might be enough to just replace 'provide evidence' with 'suggest'

Response: In accordance with the reviewer's suggestion we changed "provide evidence" with "suggest".

I'm not sure what this relates to. I think a sentence before to explain what was done to the stimuli after the FWHR measurement.

Response: Line 224, for a better clarity we rephrased the sentence as follow:

The FWHR value obtained for each face was then regressed against monkeys' viewing preferences for the same face.

Here and throughout, this phrasing is awkward. I would prefer a simpler and more accurate wording: 'was not significant'.

Response: Line 277. We replaced "failed to reach significance" with "was not significant".

Couldn't this be checked by comparing the looking time outside the ROI? If so, it would be worth adding.

Response: We have performed a model comparison analysis on the entire exploration density over the screen. Please see supplementary Fig. 1 and point 2 of Reviewer 1.

P 14. I think this is where pupil dilation could be useful. If pupil dilation, and therefore arousal, is higher when looking at untrustworthy faces this could suggest that macaques avoided them. If pupil dilation was higher for trustworthy faces this would strengthen the approach

Response: We thank the reviewer for suggesting pupil size analysis which we could perform only in humans. We did not find a difference in pupil dilation between the two face stimuli. We observed that subjects constricted more their pupil when looking at the faces compared to the rest of the screen, which we believe it may be linked to a low luminance levels difference plus possible enhanced visual attention on faces.

Here and throughout, I think more widely used terms such as 'honest signal' or 'reliable cue' could be used here. Rather than detect, it would be to communicate trustworthiness.

Response: We changed "objective feature" with "reliable cue" and "detect" with "communicate" at line 272.

As noted by the authors slightly below, a more parsimonious explanation would be that these macaques have learned to prefer small FWHR through past interactions with caregivers and researchers, and/or that human faces with small FWHR are more similar to macaque faces which would lead to their preference. This could be mentioned as well.

Response: Line 283-284 We added this sentence:

"Finally, human faces with small FWHR may resemble most macaque faces, which would explain their preference."

I would rephrase as 'was not significant'

Response: Line 277 We replaced "failed to reach significance" with "was not significant".

'Some aspects of the personality' would be more cautious and accurate.

Response: Line 285 We now use “some aspects of the personality”

There is some evidence that nonhuman primates can use faces to inform behaviour. See for example:

*Waller, B. M., Whitehouse, J., & Micheletta, J. (2016). Macaques can predict social outcomes from facial expressions. *Animal cognition*, 19(5), 1031-1036. Buttelmann, D., Call, J., & Tomasello, M. (2009). Do great apes use emotional expressions to infer desires?. *Developmental science*, 12(5), 688-698. Morimoto, Y., & Fujita, K. (2011). Capuchin monkeys (*Cebus apella*) modify their own behaviors according to a conspecific's emotional expressions. *Primates*, 52(3), 279-286.*

Response: Line 309-310. Citation of these studies were inserted

REVIEWERS' COMMENTS:

Reviewer #1 (Remarks to the Author):

The authors have done a great job in revising the manuscript, and I appreciate their effort in collecting a new group of human subjects. The manuscript is substantially improved – clearer and more convincing. I have just a few minor concerns that I would like to see addressed.

- 1) The authors have added individual data points to Fig 1A and 1D – could they also add lines connecting the points for the same subjects. This will help show the consistency across subjects and more closely matches the statistical analysis, which is a paired test.
- 2) For the analysis of the fixation sequence in humans, why were the first three used rather than two as in monkeys?
- 3) I appreciate that the correlation analysis with age was conducted in response to one of the other reviewers, but with such a small sample, the results need to be taken with a grain of salt. I think it would be good to add an explicit caveat about the sample size and exploratory nature of the analysis in the text to avoid any readers over-interpreting these results (especially given recent concerns in the literature over sample sizes for correlations).

Reviewer #2 (Remarks to the Author):

The authors have addressed my concerns.

Reviewer #3 (Remarks to the Author):

I am satisfied with the responses made to my comments and resulting changes. The authors have done a great job addressing all of the reviewers comments, including running additional experiments and analyses. The changes made to the introduction and discussion also improved the overall quality of the manuscript. I'm happy to recommend this manuscript for publication.

REVIEWERS' COMMENTS:

Reviewer #1 (Remarks to the Author):

The authors have done a great job in revising the manuscript, and I appreciate their effort in collecting a new group of human subjects. The manuscript is substantially improved – clearer and more convincing. I have just a few minor concerns that I would like to see addressed.

1) The authors have added individual data points to Fig 1A and 1D – could they also add lines connecting the points for the same subjects. This will help show the consistency across subjects and more closely matches the statistical analysis, which is a paired test.

R1. We thank the reviewer for his positive assessment of our revised manuscript. In accordance with her/his suggestion, we changed Figure 1A and 1D. We have added lines connecting individual subjects' mean looking times to trustworthy and untrustworthy faces.

2) For the analysis of the fixation sequence in humans, why were the first three used rather than two as in monkeys?

R2. For both humans and monkeys, we decided to perform the analysis when a sufficient amount of data was available across trials and subjects. Monkeys rarely performed more than two fixations on each face whereas humans made about three fixations per face.

However, the conclusions of the analysis regarding conditions in humans are similar if we consider only two fixations. We now include the analysis with two saccades instead of three (see page 9).

“There was no main effect of the face category ($F(1,19) = 0.009$; $p = 0.92$), and no significant interaction between face category and fixation order ($F(1,19) = 0.76$; $p = 0.39$) and no main effect of fixation order ($F(1,19) = 0.69$, $p = 0.41$).”

3) I appreciate that the correlation analysis with age was conducted in response to one of the other reviewers, but with such a small sample, the results need to be taken with a grain of salt. I think it would be good to add an explicit caveat about the sample size and exploratory nature of the analysis in the text to avoid any readers over-interpreting these results (especially given recent concerns in the literature over sample sizes for correlations).

R3. We felt that the correlation result was sufficiently interesting to be reported for future studies. Nevertheless, we agree with the Reviewer that it should be associated with a note of caution. We now mention the limitation of the sample size in the result section and in the discussion.

On p. 10:

Correlation between monkeys' age and trust bias. It is reasonable to assume that preference toward trustworthy- associated facial cues is shaped by experience. *Therefore, we conducted an exploratory analysis where age* was selected as an indicator of monkey's expertise in interacting with humans and correlated to the percentage of bias toward trustworthy-associated facial cues. The overall correlation did not reach significance ($r(8) = 0.365$ unilateral, $p = 0.187$), but when we excluded the only female outlier of the group, we found a positive correlation between age and preference for trustworthy-associated facial cues ($r(7) = 0.675$, unilateral, $p = 0.048$, without monkey Z), see Supplementary Fig.2.

On p. 14:

“The exploration of correlation between age and the preference toward trustworthy-associated facial cues suggest that experience may also be responsible for the expression of this bias, *though, as a note of caution, this needs to be confirmed with a larger group directly examining the effect of age.*”